# A General Framework for Producing Interpretable Semantic Text Embeddings

**Yiqun Sun, Qiang Huang,**[*] **Yixuan Tang & Anthony K. H. Tung**
School of Computing, National University of Singapore
Singapore
`{sunyq, huangq, yixuan, atung}@comp.nus.edu.sg`

**Jun Yu**
School of Intelligence Science and Engineering, Harbin Institute of Technology (Shenzhen)
Shenzhen, China
`yujun@hit.edu.cn`

## Abstract

Semantic text embedding is essential to many tasks in Natural Language Processing (NLP). While black-box models are capable of generating high-quality embeddings, their lack of interpretability limits their use in tasks that demand transparency. Recent approaches have improved interpretability by leveraging domain-expert-crafted or LLM-generated questions, but these methods rely heavily on expert input or well-prompt design, which restricts their generalizability and ability to generate discriminative questions across a wide range of tasks. To address these challenges, we introduce **CQG-MBQA** (Contrastive Question Generation - Multi-task Binary Question Answering), a general framework for producing interpretable semantic text embeddings across diverse tasks. Our framework systematically generates highly discriminative, low cognitive load yes/no questions through the **CQG** method and answers them efficiently with the **MBQA** model, resulting in interpretable embeddings in a cost-effective manner. We validate the effectiveness and interpretability of **CQG-MBQA** through extensive experiments and ablation studies, demonstrating that it delivers embedding quality comparable to many advanced black-box models while maintaining inherently interpretability. Additionally, **CQG-MBQA** outperforms other interpretable text embedding methods across various downstream tasks. The source code is available at `https://github.com/dukesun99/CQG-MBQA`.

## 1 Introduction

Text embedding is a cornerstone of Natural Language Processing (NLP), transforming texts—whether sentences, paragraphs, or full documents—into embedding vectors that capture their semantic meaning. In semantic embedding spaces, the similarity between texts is represented by the proximity of their embedding vectors, typically measured using distance measures like Euclidean distance, cosine distance, or inner product. The closer the vectors, the more semantically similar the texts. These embeddings are foundational to many downstream NLP tasks, including Semantic Textual Similarity (STS) (Agirre et al., 2012; 2013), Information Retrieval (Karpukhin et al., 2020; Thakur et al., 2021), Clustering (Aggarwal & Zhai, 2012), and more recently, Retrieval Augmented Generation (RAG) (Lewis et al., 2020; Guu et al., 2020; Asai et al., 2024).

Black-box text embedding methods, such as **Sentence-BERT** (Reimers & Gurevych, 2019), **SimCSE** (Gao et al., 2021), **WhitenedCSE** (Zhuo et al., 2023), and **AnglE** (Li & Li, 2024), excel at generating high-quality embeddings by training on vast amounts of data. These models are highly effective at capturing semantic similarities, making them indispensable for a variety of NLP tasks (Muennighoff et al., 2023). However, their black-box nature leaves the embeddings opaque to human users. These

---

[*]Qiang Huang is the corresponding author.

models do not provide insight into why certain texts are deemed similar, which becomes problematic for tasks that require transparency, especially in applications involving high-stakes decision-making, such as legal and medical domains, or in cases requiring explanations for regulatory compliance.

Interpretability in machine learning is the ability of humans to understand the reasoning behind a model's results (Miller, 2019), which is essential not only for building trust and ensuring safety but also for detecting biases and debugging models (Molnar, 2022). Recent advances have enhanced interpretability by leveraging inherently interpretable models such as **Decision Tree** (Breiman et al., 1984) and **Generalized Additive Models** (Hastie & Tibshirani, 1986), as well as model-agnostic methods like **LIME** (Ribeiro et al., 2016) and **SHAP** (Lundberg & Lee, 2017). However, these interpretable approaches lose effectiveness when applied on top of non-interpretable features generated by black-box text embedding models. Consequently, the challenge remains: how can we create interpretable text embeddings without sacrificing performance?

Recent efforts have sought to address the challenge of creating interpretable embeddings by using questions as interpretable dimensions. For instance, **ChiLL** (McInerney et al., 2023) employs yes/no questions crafted by domain experts to classify patient clinical notes, but its reliance on costly expert annotation limits its generalizability to different datasets. **QAEmb** (Benara et al., 2024) advances this concept by using task-specific prompts with examples to automatically generate yes/no questions via Large Language Models (LLMs), achieving notable success in the fMRI prediction task (Huth et al., 2016; LeBel et al., 2023; Tang et al., 2023).

Nonetheless, **QAEmb** requires meticulously crafted prompts and uses six distinct prompt templates to generate questions for the fMRI prediction task, which complicates its usage in general settings due to the need for prompt engineering expertise. Furthermore, this example-based question generation approach often produces generic, less discriminative questions, limiting its effectiveness in broader applications. Given the importance of question quality in creating effective interpretable embeddings, there is a pressing need for a systematic approach that can automatically generate meaningful and discriminative questions across various text embedding tasks.

To address this gap, we introduce **CQG-MBQA** (Contrastive Question Generation - Multi-task Binary Question Answering), a *general* framework for producing interpretable semantic text embedding, which matches the performance of many black-box models and surpasses existing interpretable baselines across various text embedding tasks. **CQG-MBQA** harnesses contrastive learning principles to prompt LLMs to generate highly discriminative binary yes/no questions, which form the dimensions of the embedding space. These questions not only capture the semantic nuances between texts but also offer human-interpretable insights. The main contributions of this work are as follows:

- We propose **CQG-MBQA**, the first general framework that tackles the challenge of generating interpretable text embeddings for a broad range of tasks, offering a practical and scalable solution for text representation.
- Our **Contrastive Question Generation (CQG)** method produces highly discriminative questions that offer high interpretability while minimizing cognitive load for users, ensuring that the semantic relationships between texts can be easily understood.
- The **Multi-task Binary Question Answering (MBQA)** model processes these binary questions efficiently at scale, significantly reducing the inference costs typically associated with LLMs, making the framework cost-effective for real-world applications.
- We validate the efficacy of **CQG-MBQA** through extensive experiments, demonstrating its robustness and practical applicability across multiple benchmarks and downstream tasks.

## 2    RELATED WORK

Text embedding is a core NLP task that transforms texts into vector representations that capture their semantic meanings. Generally, it is categorized into black-box and interpretable embeddings.

**Black-box Embedding.** Early methods for text embedding, such as **GloVe** (Pennington et al., 2014) and **Word2Vec** (Mikolov et al., 2013), typically pool word embeddings to create low-dimensional semantic representations. However, these methods, which rely on individual word embeddings, often fail to capture the full context of a text. For example, the sentences "*Most people in the world **like** Apple.*" and "*Most people in the world **do not like** Apple.*" share high lexical overlap but have opposite meanings, highlighting the limitations of such methods, which struggle to capture deeper semantic differences beyond surface-level word similarity.

To produce context-aware text embeddings, **Universal Sentence Encoder (USE)** (Cer et al., 2018) employs a transformer model (Vaswani et al., 2017) trained on a combination of unsupervised tasks and supervised fine-tuning using the Stanford Natural Language Inference (SNLI) corpus (Bowman et al., 2015). **BERT** (Devlin et al., 2019), a transformer network pre-trained on large-scale unlabeled text, can generate sentence embeddings by pooling its output representations. Subsequent models have further refined **BERT** using contrastive learning and other semantic-related objectives. For instance, **Sentence-BERT (SBERT)** (Reimers & Gurevych, 2019) pioneers the Siamese network structure for Semantic Textual Similarity (STS), while **SimCSE** (Gao et al., 2021) develop a contrastive learning framework for both unsupervised and supervised settings. **WhitenedCSE** (Zhuo et al., 2023) enhances embedding uniformity and alignment with shuffled group whitening, and **AnglE** (Li & Li, 2024) optimizes angle differences to overcome cosine similarity limitations. Despite these advances, black-box models produce embeddings that are opaque and difficult to interpret. In this work, we target generating interpretable dimensions for text embedding.

**Interpretable Embedding.** The challenge of creating interpretable embeddings has been persisted, especially with the rise of dense word embeddings. Early efforts focus on transforming word embeddings to improve interpretability. Jha et al. (2018) apply categorical knowledge in the biomedical domain to convert pre-trained embeddings into interpretable dimensions, while Senel et al. (2018) quantify interpretability by analyzing latent semantic structures. Models like **SPINE** (Subramanian et al., 2018) employ auto-encoders to create interpretable embeddings from non-interpretable ones like **GloVe** (Pennington et al., 2014) and **Word2Vec** (Mikolov et al., 2013), and **Word2Sense** (Panigrahi et al., 2019) creates interpretable dimensions based on specific word senses.

Despite progress, developing context-aware, interpretable dimensions remains difficult. Recent research has shifted towards indirectly understanding embedding spaces. For instance, Lee et al. (2022) introduce token pair contribution heatmaps to enhance interpretability in sentence similarity. Opitz & Frank (2022) introduce S$^3$BERT, which trains interpretable text embeddings by structuring SBERT embeddings into explainable subspaces aligned with Abstract Meaning Representation metrics. And Simhi & Markovitch (2023) propose transforming embedding spaces into comprehensible conceptual representations. Recent advancements like **ChiLL** (McInerney et al., 2023) generate interpretable binary features from health records by querying pre-trained LLMs with *expert-crafted* yes/no questions. LISA (Patel et al., 2023) learns interpretable text style embedding by leveraging LLMs for text style analysis and summarization, training a smaller model for efficiency and applying post-processing to refine style attributes. **QAEmb** (Benara et al., 2024) extends this by prompting LLMs to *automatically* generate questions using examples of texts and questions, demonstrating its efficacy in the fMRI prediction task. Inspired by **QAEmb**, we propose a cost-effective framework that generates high-quality questions and uses them as interpretable dimensions for text embedding.

## 3 INTERPRETABLE TEXT EMBEDDING FRAMEWORK

We present **CQG-MBQA**, an interpretable text embedding framework that uses yes/no questions as semantic dimensions. By generating a single set of versatile questions, it serves as a general-purpose solution for embedding text across diverse tasks and datasets, akin to pre-trained encoders. The answers to these questions form an interpretable embedding vector, capturing the text's core semantics. For instance, consider three questions: "*Is the article about AI?*", "*Is the article about sports?*", and "*Is the article about food?*". For the text "*Apple is a technology company.*", querying an LLM yields the answers: ["*yes*", "*no*", "*no*"], resulting in an embedding vector of $[1, 0, 0]$, which reflects the text's key features. Applying this same set of questions across all texts in a corpus produces a unified embedding matrix that encodes the semantic information of the entire dataset.

As shown in Figure 1, the **CQG-MBQA** framework consists of two phases: question generation and question answering. To generate high-quality, discriminative questions, we develop a method called **Contrastive Question Generation (CQG)**, which harnesses pre-trained dense text embedding models and generative LLMs for question generation. Details of this method are outlined in Section 3.1. Once the questions are generated, their corresponding answers form the text's embedding vector. Yet, generating answers through LLMs at scale is both time-consuming and expensive. To address this, we propose a **Multi-task Binary Question Answering (MBQA)** model. Trained with a multi-task binary classification objective, this model can generate interpretable embeddings efficiently, requiring far fewer LLM API calls. Further details on this model are provided in Section 3.2.

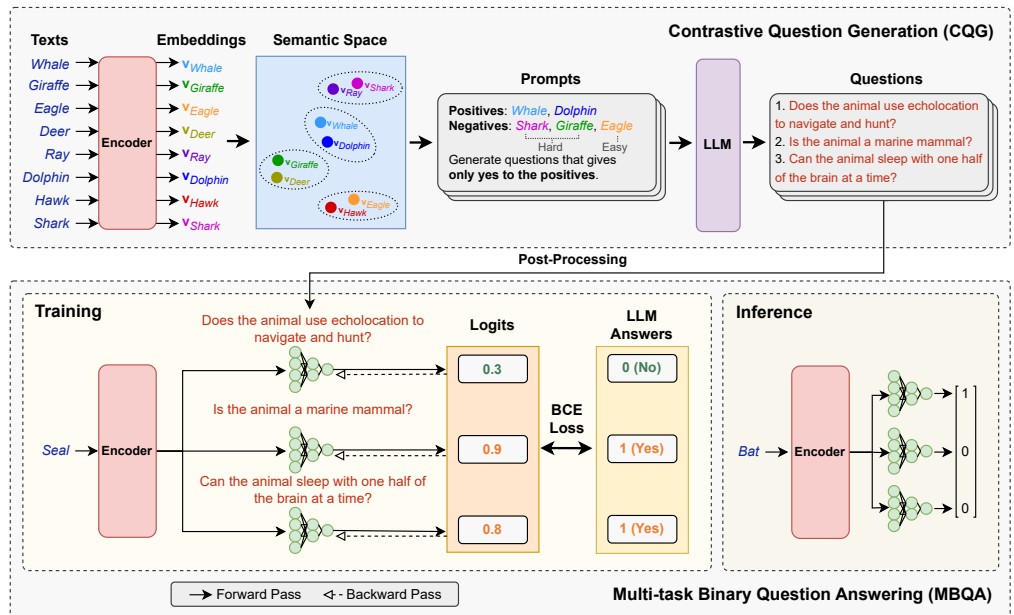

Figure 1: An overview of the **CQG-MBQA** framework.

## 3.1 QUESTION GENERATION

**Motivations.** Effective text representation with binary question answers requires highly discriminative questions to capture subtle semantic differences within the corpus. Existing methods, such as **QAEmb** (Benara et al., 2024), generate questions by prompting LLMs with dataset descriptions, example texts, and sample questions. However, this example-based approach presents two significant limitations:

(1) **Insufficient Specificity:** The generated questions are often too generic, resulting in embeddings that fail to capture fine semantic nuances.

(2) **Interpretability Issue:** A significant proportion of questions consistently yield simple "*yes*" answers, which can make the resulting embeddings more challenging to interpret and analyze.

These limitations reduce the effectiveness of the embeddings in capturing fine-grained semantic differences, which in turn hinders their practical utility in downstream tasks. To overcome these challenges, we introduce Contrastive Question Generation (CQG), a novel method that leverages the creative potential of LLMs to generate more discriminative questions.

**Contrastive Question Generation (CQG).** CQG applies contrastive learning principles, using *positive*, *hard negative*, and *easy negative* samples to guide LLMs in generating high-quality questions. These questions are designed to effectively differentiate positive samples from negative ones, especially hard negatives, which are semantically similar (Robinson et al., 2021). The goal is to generate questions that elicit a "*yes*" answer only for a specific group of texts while excluding others, even those that are closely related.

**Example 1:** Consider a toy example with four groups of texts for animals: $\mathbb{G}_1 = \{$*Whale, Dolphin*$\}$, $\mathbb{G}_2 = \{$*Shark, Ray*$\}$, $\mathbb{G}_3 = \{$*Giraffe, Deer*$\}$, and $\mathbb{G}_4 = \{$*Eagle, Hawk*$\}$. The goal is to generate questions that effectively distinguish $\mathbb{G}_1$ from other groups. At first, broad questions such as "*Does it live in water?*" or "*Is it a mammal?*" might seem useful. However, these questions are too general, as they apply to multiple groups: *Shark* and *Ray* also live in water, and *Giraffe* and *Deer* are mammals. A more precise question would be "*Does the animal use echolocation to navigate and hunt?*", which yields a "*yes*" only for *Whale* and *Dolphin*, effectively distinguishing $\mathbb{G}_1$ from the other groups. Notably, this question also generalizes to *unseen examples*, such as *Bat*, highlighting the potential of this method to generate discriminative questions applicable *beyond* the training set. △

As depicted in Figure 2, the CQG method begins by identifying semantically similar groups of texts. This is accomplished by encoding the text corpus into embedding vectors and clustering these vectors to form distinct groups. For each cluster, we design a strategic sampling technique: selecting $n_p$ positive texts from within the cluster, $n_h$ hard negative samples from neighboring clusters, and $n_e$

Figure 2: Illustration of the CQG method.

easy negative samples from the global corpus. The LLM is then prompted to generate questions under a key constraint: the questions must elicit "*yes*" answers exclusively for the positive samples and "*no*" answers for all negative samples. The detailed prompt for CQG is provided in Appendix A.1. This strategic sampling technique serves two main purposes:

(1) **Discriminative Power:** Contrasting positive samples with hard negative samples encourages the LLM to generate highly discriminative questions tailored to each cluster.
(2) **Broader Relevance:** Including easy negative samples ensures that the generated questions maintain broader relevance across the entire corpus.

A well-tuned negative-to-positive ratio, i.e., $(n_h + n_e)/n_p$, encourages the LLM to craft precise, discriminative questions for the positive cluster, leading to sparser and more interpretable embedding dimensions. This process is repeated across all clusters, resulting in a comprehensive set of LLM-generated questions that capture the unique characteristics of each cluster while maintaining global relevance, forming the foundation of our interpretable text embedding framework.

**Post-Processing.** LLMs may encounter two common issues when generating questions: (1) failing to consistently provide "*yes*" answers only for positive samples and (2) generating similar questions across different clusters. To address these challenges, we implement a post-processing step to filter and select the highest-quality, non-redundant questions.

We introduce a probing mechanism to evaluate and refine the generated questions. For each question, we randomly sample $p_p$ positive probes from the originating cluster, $p_h$ hard negative probes from neighboring clusters, and $p_e$ easy negative probes from other clusters. The LLM answers these questions, and we calculate the quality of each question using the following formula:

$$\text{quality} = \frac{\text{\# "\textit{yes}" for positive probes}}{p_p} - \frac{\text{\# "\textit{yes}" for negative probes}}{p_h + p_e}. \quad (1)$$

Equation 1 measures the difference between the percentage of "*yes*" answers for positive probes and negative probes. A higher quality value indicates that the question is more discriminative for the cluster, as it correctly identifies more positive samples while filtering out negatives.

To construct the final set of questions, we iteratively select the top $t$ most discriminative questions from each cluster, based on their quality values, and ensure that no two questions are highly semantically similar. Here, the similarity between questions is measured using cosine similarity between their corresponding embedding vectors, which are generated using the same pre-trained encoding model used in encoding the text corpus. If the cosine similarity of two questions exceeds a predefined threshold $\theta$, they are considered duplicates, and the latter question is excluded.

This post-processing step helps filter out hallucinated and/or low-quality questions generated by the LLM. The final set comprises $m$ questions, forming a diverse and highly discriminative collection that effectively captures the semantic structure of the entire corpus.

## 3.2 QUESTION ANSWERING

**Motivations.** Generating answers to questions using LLMs can be prohibitively expensive, especially when scaling up to large datasets with numerous questions. For example, as presented in Table 7, LLM-based Question Answering (QA), which leverages LLMs to answer 10,000 questions for approximately 8.8 million articles in the **MS MARCO** dev set, requires about 4.4 billion LLM inference passes and processes 1.5 trillion tokens. This incurs a substantial cost of **244,551 USD**, even with a cost-effective model (GPT-4o-mini) and a token-efficient prompting approach (grouping 20 questions per prompt). Further details are available in Appendix C.

While increasing the number of questions improves performance (see Section 4.5), the cost associated with LLM-based QA renders it impractical for large-scale real-world applications. To address this, we propose the Multi-task Binary Question Answering (MBQA) model as a cost-effective alternative.

**Multi-task Binary Question Answering (MBQA) Model.** Similar to LISA (Patel et al., 2023) and QAEmb (Benara et al., 2024) that distill the LLM answers into a smaller model, we propose the MBQA model that enables efficient inference with a single forward pass of the encoding model. It is designed to leverage LLM-generated answers from a smaller subset of texts to train a multi-task binary classification model. This model consists of a pre-trained encoding model and multiple classification heads. The encoder converts the input text into an embedding vector, while the classification heads predict binary scores for each question. Formally, the MBQA model $\mathcal{M}$ is defined as:

$$\mathcal{M} = (Enc, \{C_1, C_2, \cdots, C_m\}), \tag{2}$$

where $Enc : \mathcal{T} \to \mathbb{R}^d$ represents the encoding model, and $C_i : \mathbb{R}^d \to [0, 1]$ is the $i$-th Multi-Layer Perceptron (MLP) classification head. For a given input text $\boldsymbol{t} \in \mathcal{T}$, the MBQA model generates a binary embedding vector $\boldsymbol{y} = [y_1, y_2, \cdots, y_m] \in \{0, 1\}^m$ as follows:

$$\boldsymbol{e} = Enc(\boldsymbol{t}), \tag{3}$$
$$y_i = \mathbb{1}[\sigma(C_i(\boldsymbol{e})) > \tau], \text{ for } i \in \{1, 2, \cdots, m\}, \tag{4}$$

where $\sigma$ is the Sigmoid function, $\mathbb{1}[\cdot]$ is the indicator function, and $\tau$ is the threshold for binary classification. During training, the encoder $Enc$ is frozen and only the classification heads $\{C_1, C_2, \cdots, C_m\}$ are optimized using weighted Binary Cross-Entropy (BCE) Loss (Bishop, 2006) on the available LLM-generated question-answer pairs.

**Remarks.** The MBQA model achieves 96% accuracy in reproducing LLM-generated answers for CQG questions with just a single pass through the encoding model, substantially reducing costs compared to running a pre-trained LLM for each text. Our model only requires training data from as few as 1,000 articles per question, resulting in 10 million text-question pairs for 10,000 questions, costing just 31 USD using `GPT-4o-mini`. The training process takes 36 hours, and embedding the entire **MS MARCO** dev set requires 90 hours on a single GTX 1080 Ti, which is an inexpensive GPU. Consequently, encoding the same **MS MARCO** dev set with the MBQA model costs around **41 USD**–just **0.017%** of the original cost with `GPT-4o-mini`. This model allows us to scale up the number of questions (dimensions) efficiently, providing interpretable embeddings at a fraction of the cost of LLM-based answering. In addition, the CQG pipeline is also cost-effective, which just requires 2.52 USD for question generation and 1.92 USD for probing using `GPT-4o-mini` in our experiments. For more details on training and evaluation, see Appendix B.

## 4 EXPERIMENTS

In this section, we present a comprehensive evaluation of the **CQG-MBQA** framework by addressing four essential questions aimed at understanding its performance and applicability:

- **Embedding Quality:** How well does our framework generate high-quality interpretable embeddings comparable to advanced black-box models? (Section 4.3)
- **Interpretability:** Does our framework improve the human interpretability of embeddings over existing methods? (Section 4.4)
- **Question Efficiency:** Can the CQG method generate a limited number of highly discriminative questions that maintain strong performance? (Section 4.5)
- **Flexibility:** Can the MBQA model be tuned to strike a balance between embedding quality and interpretability? (Section 4.6)

We conduct experiments on three core downstream tasks in text embedding: STS, retrieval, and clustering, allowing us to benchmark **CQG-MBQA** against both black-box and interpretable models.

### 4.1 METRICS

**Embedding Quality Measurement.** For evaluating embedding quality, we adopt the metrics that are widely used in the MTEB benchmark (Muennighoff et al., 2023) for a comprehensive comparison. For STS tasks, we use Spearman correlation (Spearman, 1904) on cosine similarity between embeddings

Table 1: STS results measured by Spearman correlation. Evaluated on seven popular datasets: **SemEval STS tasks 2012-2016 (STS12–STS16)** (Agirre et al., 2012; 2013; 2014; 2015; 2016), **STS Benchmark (STS-B)** (Cer et al., 2017), and **SICK-Relatedness (SICK-R)** (Marelli et al., 2014) using the MTEB evaluation suite (Muennighoff et al., 2023).

| Type | Model | Spearman Correlation ↑ (STS) | | | | | | | |
|------|-------|-------|-------|-------|-------|-------|-------|--------|------|
| | | STS12 | STS13 | STS14 | STS15 | STS16 | STS-B | SICK-R | Avg. |
| Black-box | **BERT** | 38.78 | 57.98 | 57.98 | 63.15 | 61.06 | 46.35 | 58.40 | 54.81 |
| | **GloVe** | 54.64 | 69.16 | 60.81 | 72.31 | 65.34 | 61.54 | 55.43 | 62.74 |
| | **USE** | 64.49 | 67.80 | 64.61 | 76.83 | 73.18 | 74.92 | 76.69 | 71.22 |
| | **SimCSE (Unsup.)** | 66.05 | 81.49 | 73.61 | 79.72 | 78.12 | 76.52 | 72.24 | 75.39 |
| | **SBERT (Ori.)** | 74.53 | 77.00 | 73.18 | 81.85 | 76.82 | 79.10 | 74.29 | 76.68 |
| | **SimCSE (Sup.)** | 75.30 | 84.67 | 80.19 | 85.40 | 80.82 | 84.25 | 68.38 | 79.86 |
| | **WhitenedCSE** | 74.65 | 85.79 | 77.49 | 84.71 | 80.33 | 81.48 | 75.34 | 79.97 |
| | **SBERT (New)** | 73.08 | 82.13 | 76.73 | 85.58 | 80.23 | 83.09 | 79.32 | 80.02 |
| | **OpenAI** | 72.84 | 86.1 | 81.15 | 88.49 | 85.08 | 83.56 | 79.00 | 82.31 |
| | **AnglE** | 79.09 | 89.62 | 85.02 | 89.51 | 86.61 | 89.06 | 82.62 | 85.93 |
| Interp. | **Bag-of-Tokens** | 44.75 | 52.06 | 54.78 | 68.65 | 60.59 | 54.85 | 57.87 | 56.22 |
| | **QAEmb-MBQA** | 59.40 | 63.19 | 57.68 | 69.29 | 63.18 | 71.33 | 72.33 | 65.20 |
| | **CQG-MBQA** | 69.21 | 80.19 | 73.91 | 80.66 | 78.30 | 82.69 | 78.21 | 77.60 |

as the metric. In retrieval tasks, we assess the performance using nDCG@10 (Wang et al., 2013). For clustering tasks, we evaluate the results using V-Measure (Rosenberg & Hirschberg, 2007).

**Interpretability Measurement.** Since both STS and retrieval tasks measure pairwise text similarity using cosine similarity, we focus on interpreting the cosine similarity scores produced by **CQG-MBQA**. With inherently interpretable dimensions, we can offer insights to users by highlighting the dimensions that contribute most to the similarity between two texts. Building on **COGAM** (Abdul et al., 2020), we suggest that interpretability should account for the **cognitive load** imposed on users. In **COGAM**, cognitive load is assessed by counting the number of visual cognitive chunks. Similarly, we measure it by the number of questions a user must consider to understand the similarity between two texts, corresponding to the dimensions where both embedding vectors have a value of 1. Formally, for any two binary embedding vectors $\boldsymbol{u} = [u_1, u_2, \cdots, u_m]$ and $\boldsymbol{v} = [v_1, v_2, \cdots, v_m]$, the cognitive load is defined as the inner product of $\boldsymbol{u}$ and $\boldsymbol{v}$:

$$\text{cognitive load} = \langle \boldsymbol{u}, \boldsymbol{v} \rangle = \sum_{i=1}^{m} u_i v_i. \tag{5}$$

To mitigate the impact of the number of dimensions ($m$) used for different models, we also report the cognitive load normalized by $m$: normalized cognitive load $= \frac{\text{cognitive load}}{m}$. Quantifying cognitive load allows us to assess the interpretability of our **CQG-MBQA** framework's embeddings. A lower value indicates that fewer dimensions are involved, making the interpretation easier to understand, thus enhancing both interpretability and user-friendliness.

## 4.2 MODELS

**Interpretable Models.** We evaluate **CQG-MBQA** against existing interpretable baselines to provide a thorough comparison. For **CQG-MBQA**, we use the **MEDI2** dataset (Muennighoff et al., 2024), a diverse text corpus, as the training data. Texts are encoded with the **AnglE** model (`UAE-Large-V1`), and the resulting embeddings are normalized. We then perform $k$-**Means clustering** (Arthur & Vassilvitskii, 2007) with $k = 5,000$. The CQG process produces 9,614 questions after probing and deduplication, forming the final embedding dimensions. The MBQA model also leverages **AnglE** as the backbone, with three-layer MLP classification heads and a hidden layer size of 8. More implementation details of **CQG-MBQA** are provided in Appendix D.1.

To make a fair comparison and highlight the benefits of CQG, we adapt **QAEmb**, originally designed for task-specific embeddings, and develop **QAEmb-MBQA**. Specifically, we utilize its example-based question generation approach (Benara et al., 2024), modifying the prompts to generate questions suitable for general text embeddings with LLMs. After deduplication, **QAEmb** produces 10,654 questions, which we integrate with our MBQA model for evaluation. Implementation Details for **QAEmb-MBQA** are provided in Appendix D.2. Additionally, we include **Bag-of-Tokens**, a simple

Table 2: Retrieval results evaluated by nDCG@10. Evaluated on seven diverse datasets: **MS MARCO** (Bajaj et al., 2016), **NewsSpectrum (NSP)** (Sun et al., 2024), **ArguAna** (Wachsmuth et al., 2018), **FiQA-2018 (FQA)** (Maia et al., 2018), **NFCorpus (NFC)** (Boteva et al., 2016), **SCIDOCS** (Cohan et al., 2020), and **SciFact** (Wadden et al., 2020). **MS MARCO** is evaluated on a 1% sample of its dev set, while **NewsSpectrum** uses news titles as queries with corresponding articles as targets. The remaining datasets are assessed using the MTEB evaluation suite.

| Type | Model | nDCG@10 ↑ (Retrieval) | | | | | | | |
|---|---|---|---|---|---|---|---|---|---|
| | | MS MARCO | NSP | ArguAna | FQA | NFC | SCIDOCS | SciFact | Avg. |
| Black-box | BERT | 16.86 | 12.48 | 28.29 | 2.19 | 4.30 | 2.82 | 13.34 | 11.47 |
| | SimCSE (Unsup.) | 44.63 | 40.05 | 38.34 | 9.84 | 9.88 | 5.50 | 25.72 | 24.85 |
| | GloVe | 44.27 | 35.15 | 36.30 | 10.09 | 13.87 | 8.04 | 29.58 | 25.33 |
| | SimCSE (Sup.) | 47.86 | 47.01 | 38.33 | 10.41 | 12.42 | 7.53 | 29.59 | 27.59 |
| | SBERT (New) | 88.74 | 69.66 | 47.13 | 37.27 | 32.25 | 21.82 | 62.64 | 51.36 |
| | AnglE | 90.43 | 81.46 | 66.15 | 44.84 | 38.65 | 22.98 | 74.07 | 59.80 |
| | OpenAI | 92.18 | 85.17 | 58.05 | 55.00 | 42.07 | 23.11 | 77.77 | 61.91 |
| Interp. | Bag-of-Tokens | 29.79 | 22.09 | 34.25 | 3.99 | 21.51 | 6.79 | 47.36 | 23.68 |
| | BM25 | 68.42 | 76.81 | 49.28 | 25.14 | 32.08 | 15.78 | 68.70 | 48.03 |
| | QAEmb-MBQA | 40.51 | 30.45 | 34.75 | 8.23 | 3.87 | 3.74 | 12.01 | 19.08 |
| | CQG-MBQA | 62.21 | 49.63 | 47.75 | 18.63 | 9.74 | 8.67 | 32.80 | 32.78 |

Table 3: Clustering results assessed by V-Measure. Evaluated on seven commonly-used datasets: **TwentyNewsgroupsClustering (TNG)**, **StackExchangeClusteringP2P (SE-P2P)**, **BiorxivClusteringP2P (BR-P2P)**, **BiorxivClusteringS2S (BR-S2S)**, **MedrxivClusteringP2P (MR-P2P)**, **MedrxivClusteringS2S (MR-S2S)**, and **RedditClusteringP2P (RD-P2P)** from the MTEB evaluation suite.

| Type | Model | V-Measure ↑ (Clustering) | | | | | | | |
|---|---|---|---|---|---|---|---|---|---|
| | | TNG | SE-P2P | BR-P2 | BR-S2S | MR-P2P | MR-S2S | RD-P2P | Avg. |
| Black-box | SimCSE (Unsup.) | 23.21 | 28.50 | 24.90 | 19.55 | 23.60 | 21.97 | 45.14 | 26.70 |
| | GloVe | 25.83 | 28.51 | 29.27 | 19.18 | 26.12 | 20.38 | 35.82 | 26.44 |
| | BERT | 23.35 | 26.55 | 30.12 | 24.77 | 26.09 | 23.60 | 43.32 | 28.26 |
| | SimCSE (Sup.) | 34.86 | 29.45 | 30.15 | 24.67 | 26.25 | 24.12 | 47.74 | 31.03 |
| | SBERT (New) | 47.47 | 33.13 | 36.99 | 33.21 | 34.25 | 32.24 | 54.80 | 38.87 |
| | AnglE | 51.72 | 36.72 | 39.38 | 37.23 | 33.22 | 31.18 | 65.35 | 42.11 |
| | OpenAI | 58.14 | 36.88 | 38.03 | 36.53 | 32.70 | 31.27 | 67.96 | 43.07 |
| Interp. | Bag-of-Tokens | 8.52 | 17.64 | 4.70 | 3.32 | 11.39 | 13.05 | 15.67 | 10.61 |
| | QAEmb-MBQA | 36.72 | 25.68 | 24.66 | 21.16 | 25.53 | 22.85 | 46.57 | 29.02 |
| | CQG-MBQA | 40.00 | 28.22 | 34.88 | 31.13 | 31.02 | 28.71 | 54.40 | 35.48 |

baseline using the BERT tokenizer for interpretable embeddings. For retrieval, we compare against the rule-based sparse retriever **BM25** (Robertson et al., 1995), implemented using BM25S (Lù, 2024).

**Black-box Models.** To benchmark the embedding quality of **CQG-MBQA**, we compare it with several advanced black-box text embedding models. These include **GloVe** (Pennington et al., 2014; Reimers & Gurevych, 2019), **USE** (Cer et al., 2018), **BERT** (Devlin et al., 2019), the **Original (Ori.)** and **Up-to-date (New)** versions of **Sentence-BERT (SBERT)** (Reimers & Gurevych, 2019), the **Supervised (Sup.)** and **Unsupervised (Unsup.)** versions of **SimCSE** (Gao et al., 2021). Additionally, we compare our framework with **WhitenedCSE** (Zhuo et al., 2023), the **OpenAI** API, and **AnglE** (Li & Li, 2024). Implementation details for all baseline models are outlined in Appendix D.2.

## 4.3 EMBEDDING QUALITY

Tables 1–3 present the embedding quality results for STS, retrieval, and clustering tasks, highlighting **CQG-MBQA**'s competitive performance. In STS tasks (Table 1), **CQG-MBQA** achieves comparable quality to advanced dense models like **SimCSE** and **SBERT (New)** while preserving interpretability. It also outperforms earlier methods like **GloVe**, **USE**, and **BERT**, as well as all interpretable baselines. For retrieval tasks (Table 2), **CQG-MBQA** surpasses **SimCSE**, **GloVe**, and **BERT**. While it trails state-of-the-art black-box models such as **AnglE** and **OpenAI**, it consistently outperforms all interpretable baselines except for **BM25**, a rule-based model optimized for retrieval. In clustering tasks (Table 3), **CQG-MBQA** exceeds several black-box models (**SimCSE**, **GloVe**, and **BERT**) and outperforms all interpretable baselines, closely matching the performance of recent models like

Table 4: Cognitive Load and Normalized Cognitive Load (values in parentheses) on STS datasets.

| Model | Cognitive Load ↓ (Normalized Cognitive Load ↓) (STS) | | | | | | | |
|---|---|---|---|---|---|---|---|---|
| | STS12 | STS13 | STS14 | STS15 | STS16 | STS-B | SICK-R | Avg. |
| Bag-of-Tokens | 8 (0.03%) | 4 (0.01%) | 6 (0.02%) | 5 (0.02%) | 8 (0.03%) | 7 (0.02%) | 6 (0.02%) | 6 (0.02%) |
| QAEmb-MBQA | 1,626 (15.26%) | 1,571 (14.75%) | 1,625 (15.25%) | 1,443 (13.54%) | 1,577 (14.80%) | 1,408 (13.22%) | 1,018 (9.56%) | 1,467 (13.77%) |
| CQG-MBQA | 481 (5.00%) | 439 (4.57%) | 458 (4.76%) | 426 (4.43%) | 478 (4.97%) | 446 (4.64%) | 413 (4.30%) | 449 (4.67%) |

| Text A | Cocoa flavanol consumption improves cognitive function, blood pressure control, and metabolic profile in elderly subjects: the Cocoa, Cognition, and Aging (CoCoA) Study—a randomized controlled trial1234 |
|---|---|
| Text B | Blood Pressure Is Reduced and Insulin Sensitivity Increased in Glucose-Intolerant, Hypertensive Subjects after 15 Days of Consuming High-Polyphenol Dark Chocolate |

| ID | Questions | Answer A | Answer B |
|---|---|---|---|
| 283 | Is the article intended for educational purposes? | Yes | Yes |
| 357 | Is the research based on data collected from human participants? | Yes | Yes |
| 1153 | Is the content related to personal health or well-being? | Yes | Yes |
| 2039 | Is there a focus on the effects of a specific disease or disorder? | No | Yes |
| 3400 | Is the research aimed at understanding or treating neurological conditions? | Yes | No |
| 4634 | Is the method discussed in the article aligned with scientific standards? | Yes | Yes |
| 7273 | Is there a discussion on the role of food in human health? | No | Yes |
| 8540 | Is the subject matter related to medical or health professions? | Yes | Yes |
| 9292 | Is there an emphasis on the importance of variables? | Yes | No |

Figure 3: Case study.

**SBERT (New)**. The comparison with **QAEmb-MBQA** further underscores the efficacy of our CQG algorithm in generating high-quality, discriminative questions that capture semantic nuances.

### 4.4 INTERPRETABILITY

**Cognitive Load.** Table 4 displays the cognitive load required to interpret embeddings across different interpretable models, evaluated through the STS task, which computes the pairwise similarity of texts. This provides a clear measure of how much effort is needed to understand the embeddings. **CQG-MBQA** achieves a 2.5∼3.6× lower cognitive load than **QAEmb-MBQA**, indicating that the CQG method significantly enhances interpretability. This technique produces more "*no*" answers and fewer "*yes*" answers, making the embeddings easier to interpret. For **Bag-of-Tokens**, it has a much lower cognitive load due to its lexical nature, but this advantage comes at the cost of significantly reduced embedding quality. The trade-off between interpretability and embedding quality can be adjusted by tuning the binary classification threshold $\tau$, as further discussed in Section 4.6.

**Case Study.** Figure 3 showcases a pair of texts from our training corpus, focusing on nine specific questions (dimensions) where at least one text yields a "*yes*" answer. This illustrates how **CQG-MBQA** generates relevant and insightful dimensions. For instance, question ID 1153, which asks if the text is related to personal health or well-being, receives a "*yes*" for both Text A and Text B, accurately reflecting their shared focus on health topics. Similarly, question ID 4634 inquires whether the text aligns with scientific standards, and both texts–discussing evidence-based findings on substance effects–obtain "*yes*" answers, showcasing the relevance of generated questions.

Despite the texts' similarity, **CQG-MBQA** captures subtle semantic differences through fine-grained questions. For example, question ID 3400 asks if the research targets neurological conditions. Text A, which discusses cognitive function, receives a "*yes*", indicating a connection to neurological conditions, whereas Text B, focusing on blood pressure and insulin sensitivity, acquires a "*no*", highlighting a clear distinction in their subject matter. This case study highlights how CQG generates interpretable, relevant, and discriminative questions that effectively capture nuanced semantic differences, while MBQA accurately predicts the answers, reinforcing the framework's practicality and reliability.

### 4.5 EFFECT ON NUMBER OF QUESTIONS

We investigate how the number of questions (dimensions) $m$ affects the quality and interpretability of **CQG-MBQA** embeddings produced by varying the length of the final binary embedding vector.

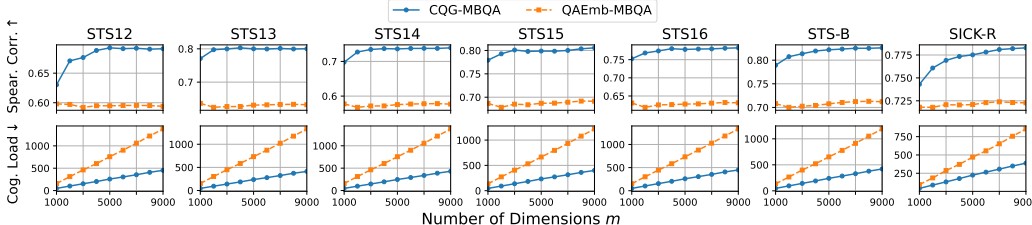

Figure 4: Spearman correlation and cognitive load vs. the number of dimensions $m$. Higher Spearman correlation signals better embedding quality; lower cognitive load implies greater interpretability.

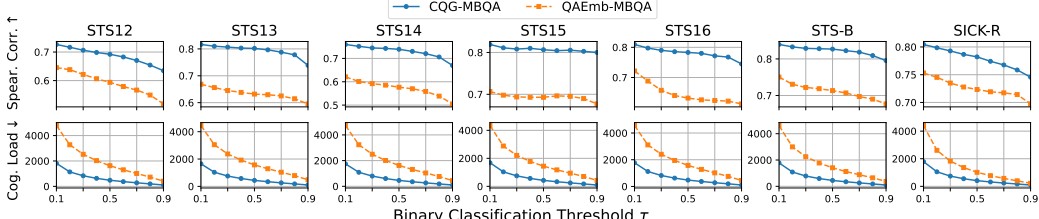

Figure 5: Spearman correlation and cognitive load vs. the binary classification threshold $\tau$.

Figure 4 shows that as $m$ grows, the Spearman correlation increases and stabilizes around 3,000 dimensions, indicating enhanced embedding quality. This highlights the need for an efficient QA model to manage computational costs while scaling up dimensions for optimal embedding quality.

However, a trade-off arises: as $m$ increases, cognitive load increases, and the interpretability declines due to a higher proportion of "*yes*" in the embeddings. This inverse relationship between embedding quality and interpretability emphasizes the importance of balancing dimensions based on the task's requirements. Figure 4 also demonstrates the effectiveness of the CQG algorithm in generating high-quality, discriminative embeddings with approximately 3,000 questions, achieving a balance between embedding quality and interpretability without an excessive number of questions.

## 4.6 TRADE-OFF BETWEEN EMBEDDING QUALITY AND INTERPRETABILITY

To further examine the trade-off between embedding quality and interpretability, we vary the binary classification threshold $\tau$ that determines the final binary embedding vector. Figure 5 illustrates this balance: increasing $\tau$ improves interpretability, but this comes at the cost of reduced embedding quality, as the Spearman correlation decreases. This occurs because fewer active dimensions simplify interpretation but may lose subtle semantic distinctions.

More importantly, this trade-off presents an opportunity for user-driven customization. Depending on different scenarios, users of our framework can dynamically tune the desired $\tau$ based on the cognitive load to meet their needs. For instance, in scenarios requiring rapid decision-making or where cognitive resources are constrained, users might prioritize interpretability by opting for a higher threshold. On the other hand, in situations where nuanced analysis is crucial and resources are abundant, a lower threshold could be chosen to maximize embedding quality. This flexibility makes **CQG-MBQA** highly adaptable to different scenarios and user requirements.

## 5 CONCLUSION

In this paper, we introduce **CQG-MBQA**, a novel general framework for generating interpretable semantic text embeddings by systematically creating binary questions and using the answers as interpretable embedding dimensions. Our CQG method effectively addresses the challenges of generalizability and quality issues during the question generation phase, while the MBQA model provides an efficient, scalable solution for answering these questions, significantly reducing costs. Through extensive experiments on STS, retrieval, and clustering tasks, we demonstrate that our framework delivers performance comparable to advanced black-box models while being inherently interpretable. Moreover, **CQG-MBQA** consistently outperforms other interpretable text embedding models across various downstream tasks, further validating its effectiveness.

ACKNOWLEDGMENTS

This research is supported by the Ministry of Education, Singapore, under its MOE AcRF TIER 3 Grant (MOE-MOET32022-0001), the National Research Foundation, Singapore under its AI Singapore Programme (AISG Award No: AISG3-RP-2022-029), and the National Natural Science Foundation of China under grant No. 62125201. Any opinions, findings, and conclusions or recommendations expressed in this material are those of the author(s) and do not reflect the views of the Ministry of Education, Singapore and the National Research Foundation, Singapore.

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

## A PROMPTS

### A.1 PROMPT: CONTRASTIVE QUESTION GENERATION

We present the prompt template used for the Contrastive Question Generation (CQG) algorithm. The input to the prompt template consists of two lists of texts: *positive_examples* and *negative_examples*. The LLM is explicitly instructed to generate questions that yield "*yes*" answers for the positive examples and "*no*" answers for the negative examples. To enhance the quality of the generated questions and the potential generalizability to new texts, we prompt the LLM to avoid complex sentence structures. Initial experiments revealed that if we do not require the LLM to avoid complex sentence structures, the LLM tended to create discriminative questions by simply combining two conditions to ensure "*yes*" answers for the positives, rather than identifying deeper relationships between the positives and negatives. Additionally, we include formatting instructions at the end of the prompt template to improve result parsing accuracy.

> Generate 10 simple yet insightful yes/no questions that determine the properties of an article, where for all questions, the answer will be "yes" for ALL the positive articles and "no" for ALL the negative articles. Keep questions concise and avoid using complex sentence structures with "and" or "or" unless necessary.
> **Positive Articles:**
> Positive {$i$}. {*positive_example_i*}
> **Negative Articles:**
> Negative {$i$}. {*negative_example_i*}
> **Instruction:** Based on the excerpts provided, generate 10 simple yet insightful yes/no questions that can accurately differentiate the positive articles from the negative articles. Each question should be concise and framed in such a way that it will elicit a "yes" response for ALL positive articles and a "no" response for ALL negative articles. Avoid using complex sentence structures with "and" or "or" unless absolutely necessary. Format the questions in a numbered list as shown below:
> 1. First simple yes/no question
> 2. Second simple yes/no question

### A.2 PROMPT: QAEMB QUESTION GENERATION

The Question Generation Prompt #5 used in the QAEmb paper (Benara et al., 2024) is the most general form prompt in the list, making it suitable for adopting it to generate questions for general-purpose text embedding. It uses two lists of texts as input: *example narrative sentences* and *example yes/no questions*. The original prompt instructs the LLM to *"Generate a bulleted list of 100 specific, non-overlapping yes/no questions that ask about aspects of the example narrative sentences that are important for classifying them."* This was originally designed for the task of fMRI prediction with narrative sentences.

Following this approach, we designed a prompt template for experiments on QAEmb question generation, also using two lists of texts as inputs: *reference_articles* and *example_questions*. The example questions are sourced from the original QAEmb paper, while the reference articles are randomly drawn from the training dataset.

> Generate 10 diverse insightful yes/no questions that determine the properties of an article.
> **Reference Articles:**
> {*i*}. {*reference_article_i*}
> **Example Questions:**
> {*i*}. {*example_question_i*}
> **Instruction:** Based on the excerpts provided, generate 10 yes/no questions that can determine the properties of the articles. Format the questions in a numbered list as shown below:
> 1. First yes/no question
> 2. Second yes/no question

### A.3 PROMPT: MULTI-TASK BINARY QUESTION ANSWERING

This section details the prompt template used to generate LLM answers for training the Multi-task Binary Question Answering (MBQA) model. The prompt takes two inputs: the *text_chunk*, which is the training article sample, and a list of questions to be answered. To optimize token usage for the article sample and instructions, we group up to 20 questions in a single prompt.

> Evaluate the following text chunk based on the yes/no questions provided.
> **Text Chunk:**
> {*text_chunk*}
> **Questions:**
> i. {*question_i*}
> **Instruction for the model:** Please read the provided text chunk and answer each of the questions with either "yes" or "no". Format the responses as follows:
> 1. yes/no
> 2. yes/no

## B TRAINING AND EVALUATION OF THE MBQA MODEL

To ensure that the MBQA model produces faithful answers to the questions, we evaluate its question-answering performance on a 10% held-out document set that was not used for training.

**Data Collection.** For each question generated in the previous question generation step, we randomly sample 500 in-cluster samples, 300 neighboring cluster samples from 5 nearest clusters, and 200 random samples from the entire corpus. We use the pre-trained LLM (GPT-4o-mini) to generate answers for each question across these samples. The LLM-generated answers are batched in groups of 20 questions to train the multi-task binary classification model. Refer to Appendix A.3 for the prompt used to collect answers. This approach allows us to gather data from a larger number of text samples, thereby increasing the generalizability of our model.

**Training.** We train the MBQA model using the Adam optimizer with a learning rate $\alpha$ of 1e-4 and a batch size of one text sample. For each step, we calculate the loss based on all available questions with answers from the previous data collection phase. The model is trained using the BCEWithLogitsLoss function, where the weight is the ratio of "*yes*" answers to "*no*" answers in the training data.[1] Specifically, for **CQG-MBQA**, we set this weight to 7.5127, and for **QAEmb-MBQA**, the weight is set to 4.9608. The model is trained for 3 million steps, at which point performance begins to converge.

**Evaluation.** The classification results (with threshold $\tau = 0.5$) on the held-out test set for **CQG-MBQA** and **QAEmb-MBQA** are presented in Tables 5 and 6, respectively. The **CQG-MBQA** model achieves an accuracy of 96% and a macro F1 score of 91%, while the **QAEmb-MBQA** model attains a high 93% accuracy and 89% macro F1 score. These results demonstrate that MBQA can accurately predict LLM-generated question answers, serving as a cost-effective substitute for the more expensive LLM model in generating embeddings.

---

[1] https://pytorch.org/docs/stable/generated/torch.nn.BCEWithLogitsLoss.html

Table 5: Question answering performance of the **CQG-MBQA** model.

| Class | Precision ↑ | Recall ↑ | F1-score ↑ | Support |
|---|---|---|---|---|
| "*no*" | 1.00 | 0.96 | 0.97 | 846,089 |
| "*yes*" | 0.74 | 0.97 | 0.84 | 112,645 |
| **Macro Avg.** | 0.87 | 0.96 | 0.91 | 958,734 |
| **Weighted Avg.** | 0.97 | 0.96 | 0.96 | 958,734 |
| **Accuracy ↑** | | 0.96 | | |

Table 6: Question answering performance of the **QAEmb-MBQA** model.

| Class | Precision ↑ | Recall ↑ | F1-score ↑ | Support |
|---|---|---|---|---|
| "*no*" | 0.99 | 0.93 | 0.96 | 886,749 |
| "*yes*" | 0.72 | 0.94 | 0.82 | 178,591 |
| **Macro Avg.** | 0.85 | 0.93 | 0.89 | 1,065,340 |
| **Weighted Avg.** | 0.94 | 0.93 | 0.93 | 1,065,340 |
| **Accuracy ↑** | | 0.93 | | |

## C  COST ANALYSIS FOR LLM-BASED QA VS. MBQA

We estimate the cost of LLM-based QA and MBQA for producing question answers for interpretable embeddings for the entire **MS MARCO** dev set.

**LLM-based QA.** Using LLMs to answer questions for document embedding is prohibitively expensive. Table 7 shows the cost of running LLM-based QA on the **MS MARCO** dev set for various numbers of questions across different models. We assume grouping 20 questions into one prompt, using the format in Appendix A.3. Using this prompt, LLM-based Question Answering (QA), which leverages LLMs to answer 10,000 questions for approximately 8.8 million articles in the **MS MARCO** dev set, requires about 4.4 billion LLM inference passes and processes 1.5 trillion tokens. Using the Batch API, the cost per 1 million tokens for `GPT-4o` is 2.5 USD for input tokens and 7.5 USD for output tokens, while for `GPT-4o-mini`, it's 0.075 USD for input tokens and 0.3 USD for output tokens.[2] Based on the prices above, this incurs a substantial cost of **244,551 USD**, even with a cost-effective model (`GPT-4o-mini`) and a token-efficient prompting approach.

**MBQA.** The cost of running our MBQA model comprises two key components: (1) LLM API cost for training data collection and (2) GPU runtime expenses. For the first component, an upfront cost of approximately 31 USD is required to collect training data using `GPT-4o-mini` on the **MEDI2** dataset. This cost covers 1,000 text-question pairs for each of 10,000 questions, resulting in a total of 10 million text-question pairs. The cost scales down proportionally for fewer questions. For the second component, we measure the training and inference time of our model on a single GTX 1080 Ti GPU. We estimate the cost based on a rental rate of 0.08 USD per hour.[3] Training for 3 million steps took around 36 hours, and inference times for the **MS MARCO** dataset varied by the number of dimensions: 48 hours for 2,000 dimensions, 63 hours for 4,000, 73 hours for 6,000, 79 hours for 8,000, and 90 hours for 10,000 dimensions.

## D  IMPLEMENTATION DETAILS

### D.1  THE CQG-MBQA FRAMEWORK

In the experiments, we train the proposed **CQG-MBQA** framework using the **MEDI2** dataset (Muennighoff et al., 2024). Detailed information about the model configuration and hyperparameters used in our framework is provided in Table 8.

---

[2]Costs obtained from `https://openai.com/api/pricing`
[3]Costs obtained from `https://vast.ai/pricing/gpu/GTX-1080-TI`

Table 7: Estimated cost for embedding the **MS MARCO** dev set using LLM-generated answers.

| Model | Cost for Number of Questions | | | | |
|---|---|---|---|---|---|
| | 2,000 | 4,000 | 6,000 | 8,000 | 10,000 |
| **GPT-4o-mini** | $48,859 | $97,792 | $146,725 | $195,570 | $244,551 |
| **GPT-4o** | $1,454,000 | $2,910,487 | $4,366,946 | $5,820,517 | $7,278,566 |
| **MBQA** | $13 | $20 | $27 | $34 | $41 |

Table 8: Hyperparameters used in our experiments.

| Description | Symbol | Setting |
|---|---|---|
| Encoding model | $Enc$ | UAE-Large-V1 |
| Generation model | $LLM$ | GPT-4o-mini-2024-07-18 |
| Number of clusters | $k$ | 5,000 |
| Positive samples per cluster | $n_p$ | 6 |
| Hard negative samples per cluster | $n_h$ | 18 |
| Easy negative samples per cluster | $n_e$ | 18 |
| Positive probe samples per question | $p_p$ | 5 |
| Hard negative probe samples per question | $p_h$ | 3 |
| Easy negatives probe samples per question | $p_e$ | 2 |
| Deduplication threshold | $\theta$ | 0.8 |
| Top questions per cluster | $t$ | 4 |
| Learning rate of the MBQA Model | $\alpha$ | 1e-4 |
| Binary classification threshold | $\tau$ | 0.5 |

**Data Pre-processing.** We use the **MEDI2** dataset, downloaded from the HuggingFace repository at `GritLM/MEDI2`.[4] We filter out files starting with *task*, as they are unsuitable for the training corpus. From the remaining files, we extract both positive and negative instances of each data line. Since the **MEDI2** dataset contains instructions for queries and documents, we remove the instruction part, leaving only the content. We merge all positive and negative instances from the filtered corpus and run a simple exact deduplication to produce the final training text corpus.

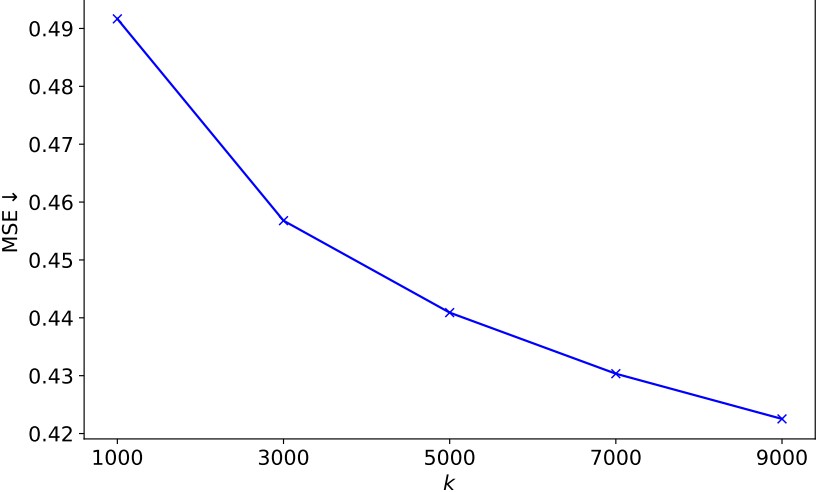

Figure 6: Mean Squared Error (MSE) vs. number of cluster $k$ for optimal $k$ selection.

**Contrastive Question Generation.** The pre-processed training corpus contains 6.8 million unique texts. We encode these texts using **AnglE** (`UAE-Large-V1`) and normalize the embeddings. We then run **KMeans** clustering (Arthur & Vassilvitskii, 2007) with $k = 5,000$ clusters and default

---

[4] https://huggingface.co/datasets/GritLM/MEDI2/tree/main

Table 9: Model checkpoints used in our experiments.

| Model | Checkpoint |
|---|---|
| **BERT** | `bert-base-uncased` |
| **GloVe** | `average_word_embeddings_glove.6B.300d` |
| **SimCSE (Unsup.)** | `unsup-simcse-bert-base-uncased` |
| **SimCSE (Sup.)** | `sup-simcse-bert-base-uncased` |
| **SBERT (New)** | `all-MiniLM-L12-v2` |
| **OpenAI** | `text-embedding-ada-002` |
| **AnglE** | `UAE-Large-V1` |
| **BM25** | `bm25s` |

parameters, utilizing using the scikit-learn library,[5] accelerated by Intel(R) Extension for scikit-learn.[6] We pick $k = 5,000$ based on the elbow point observed in the MSE vs. the number of clusters plotted in Figure 6. Once clustering is complete, we generate questions for each cluster according to the process described in Section 3.1. We sample random positive examples from within the cluster, hard negatives from the three nearest clusters, and easy negatives from the remaining clusters. This process yields 9,614 deduplicated questions, which serve as the final embedding dimensions.

**Multi-Task Binary Question Answering.** We train the MBQA model following the setup described in Appendix B.

### D.2 BASELINE MODELS

**QAEmb-MBQA.** To ensure a fair comparison with **QAEmb** (Benara et al., 2024), we adapt **QAEmb** and develop **QAEmb-MBQA** by utilizing its example-based prompting method. Specifically, we use the prompts originally designed for the fMRI task and modify them to create a prompt suitable for generating a list of questions for the general text embedding task. Detailed prompts used for this version are provided in Appendix A.2. Using this prompt, we generate questions 5,000 times, each time utilizing $n_p + n_h + n_e = 42$ randomly sampled documents from the training corpus as reference articles. The generated questions are deduplicated using a process similar to that of the CQG method, where questions with a cosine similarity score higher than $0.925$ (based on question embeddings) are removed. The QAEmb question generation process resulted in a total of $10,654$ unique questions. Finally, the MBQA model is trained using the same approach as in our **CQG-MBQA** framework.

**Black-box Models.** For STS tasks, the results for **SBERT (Ori.)** and **USE** are sourced from (Reimers & Gurevych, 2019), and the results for **WhitenedCSE** are taken from the best-performing model in (Zhuo et al., 2023), all evaluated using the same metric. Table 9 shows the model checkpoints used for each black-box model. For STS, retrieval (excluding **MS MARCO** and **NewsSpectrum**), and clustering tasks, the results for **BERT**, **GloVe**, **SimCSE (Unsup.)**, **SimCSE (Sup.)**, **SBERT (New)**, **OpenAI**, and **AnglE** are retrieved from the MTEB leaderboard.[7] The retrieval results for **BM25** (excluding the datasets **MS MARCO** and **NewsSpectrum**) are also retrieved from the MTEB leaderboard, while our own experiments were conducted for **MS MARCO** and **NewsSpectrum**.

## E ADDITIONAL DOWNSTREAM TASKS

In addition to the STS, retrieval, and clustering tasks presented in Section 4, we further evaluate our framework on four additional downstream tasks from the MTEB benchmark: classification, pair classification, reranking, and summarization.[8] The classification results are provided in Table 10, while results for the pair classification, reranking, and summarization tasks are detailed in Table 11.

---

[5] https://scikit-learn.org/stable/modules/generated/sklearn.cluster.KMeans.html
[6] https://github.com/intel/scikit-learn-intelex
[7] https://huggingface.co/spaces/mteb/leaderboard
[8] The bitext mining task from the MTEB benchmark is excluded because it involves multilingual inputs, whereas our framework currently focuses on single-language scenarios, which are beyond the scope of this study.

Table 10: Classification results, measured by Accuracy, on twelve datasets: **AmazonCounterfactual (AC)**, **AmazonPolarity (AP)**, **AmazonReviews (AR)**, **Banking77 (Bank)**, **Emotion (Emo)**, **Imdb**, **MassiveIntent (MaI)**, **MassiveScenario (MaS)**, **MTOPDomain (MTD)**, **MTOPIntent (MTI)**, **ToxicConversations (TC)**, and **TweetSentimentExtraction (TSE)**, all from the MTEB evaluation suite (Muennighoff et al., 2023).

| Type | Model | Accuracy ↑ (Classification) | | | | | | | | | | | | |
|---|---|---|---|---|---|---|---|---|---|---|---|---|---|---|
| | | AC | AP | AR | Bank | Emo | Imdb | MaI | MaS | MTD | MTI | TC | TSE | Avg. |
| Black-box | **BERT** | 74.25 | 71.33 | 33.56 | 63.41 | 35.28 | 65.35 | 59.88 | 64.28 | 82.63 | 68.14 | 70 | 51.81 | 61.66 |
| | **SimCSE (Unsup.)** | 67.09 | 74.48 | 33.85 | 73.55 | 42.22 | 69.63 | 59.84 | 66.25 | 81.71 | 59.23 | 68.82 | 53.36 | 62.50 |
| | **GloVe** | 56.91 | 60.32 | 29.67 | 67.69 | 36.93 | 62.57 | 56.19 | 66.03 | 79.11 | 55.85 | 65.4 | 50.8 | 57.29 |
| | **SimCSE (Sup.)** | 75.75 | 82.47 | 39.6 | 75.76 | 44.81 | 73.53 | 65.95 | 70.78 | 84.29 | 63.14 | 72.04 | 59.73 | 67.32 |
| | **SBERT (New)** | 65.28 | 62.98 | 30.79 | 80.4 | 41.17 | 59.76 | 67.15 | 74.58 | 91.9 | 62.84 | 67.47 | 54.25 | 63.21 |
| | **AnglE** | 75.55 | 92.84 | 48.29 | 87.69 | 51.75 | 92.78 | 76.5 | 79.75 | 94.02 | 76.92 | 71.09 | 59.75 | 75.58 |
| | **OpenAI** | 75.94 | 86.72 | 44.78 | 80.66 | 48.74 | 77.98 | 70.15 | 75.33 | 92.13 | 64.68 | 72.29 | 61.81 | 70.93 |
| Interp. | **Bag-of-Tokens** | 78.87 | 55.28 | 27.95 | 60.63 | 22.17 | 53.32 | 48.79 | 49.63 | 72.77 | 58.41 | 53.24 | 43.59 | 52.05 |
| | **QAEmb-MBQA** | 59.81 | 84.43 | 40.31 | 77.72 | 39.68 | 89.27 | 62.52 | 68.87 | 80.95 | 60.23 | 59.91 | 56.03 | 64.98 |
| | **CQG-MBQA** | 62.62 | 93.66 | 45.39 | 83.45 | 46.04 | 92.8 | 70.2 | 74.9 | 89.79 | 66.95 | 60.79 | 61.48 | 70.67 |

Table 11: Pair classification results, measured by Precision, on three datasets: **SprintDuplicateQuestions (SDQ)**, **TwitterSemEval2015 (TSE)**, and **TwitterURLCorpus (TUC)**; Reranking results, measured by MAP, on four datasets: **AskUbuntuDupQuestions (AUDP)**, **MindSmall (MS)**, **SciDocsRR (SDRR)**, and **StackOverflowDupQuestions (SODQ)**; Summarization results, measured by Spearman Correlation, on one dataset: **SummEval**, all from the MTEB evaluation suite (Muennighoff et al., 2023).

| Type | Model | Precision ↑ (Pair Classification) | | | | MAP ↑ (Reranking) | | | | | Spear. Corr. ↑ (Sum.) |
|---|---|---|---|---|---|---|---|---|---|---|---|
| | | SDQ | TSE | TUC | Avg. | AUDP | MS | SDRR | SODQ | Avg. | SummEval |
| Black-box | **BERT** | 36.81 | 55.9 | 76.29 | 56.33 | 45.84 | 28.37 | 64.94 | 34.62 | 43.44 | 29.82 |
| | **SimCSE (Unsup.)** | 78.03 | 61.01 | 81.37 | 73.47 | 51.57 | 28.62 | 66.33 | 39.35 | 46.47 | 31.15 |
| | **GloVe** | 86.96 | 53.12 | 77.35 | 72.48 | 49.57 | 27.01 | 62.56 | 34.03 | 43.29 | 28.87 |
| | **SimCSE (Sup.)** | 73.04 | 67.75 | 83.89 | 74.89 | 51.8 | 29.3 | 70.14 | 38.9 | 47.54 | 31.17 |
| | **SBERT (New)** | 92.58 | 70.02 | 84.77 | 82.46 | 64.06 | 31.02 | 87.2 | 51.47 | 58.44 | 27.9 |
| | **AnglE** | 97.24 | 78.17 | 86.33 | 87.25 | 64.2 | 32.51 | 87.49 | 55.32 | 59.88 | 32.03 |
| | **OpenAI** | 92.17 | 75.28 | 87.22 | 84.89 | 62.05 | 31.45 | 81.22 | 50.54 | 56.32 | 30.8 |
| Interp. | **Bag-of-Tokens** | 83.33 | 59.82 | 78.63 | 73.26 | 49.28 | 23.99 | 56.2 | 37.99 | 41.86 | 28.2 |
| | **QAEmb-MBQA** | 43.71 | 60.04 | 73.21 | 59.65 | 54.7 | 28.73 | 70.86 | 40.81 | 48.78 | 28.57 |
| | **CQG-MBQA** | 81.77 | 67.42 | 79.13 | 76.11 | 59.61 | 30.83 | 81.72 | 47.33 | 54.87 | 30.41 |

As shown in Tables 10 and 11, **CQG-MBQA** consistently outperforms existing interpretable text embedding models and achieves results comparable to many advanced black-box models across all examined downstream tasks. These findings underscore the framework's ability to maintain a balance between interpretability and embedding quality. Furthermore, the robust performance across diverse text embedding tasks highlights its generalizability, offering a compelling solution for tasks requiring both transparency and high semantic fidelity.

# F  ABLATION STUDIES

## F.1  COMPONENTS IN THE CQG METHOD

We perform extensive ablation studies to analyze the contributions of various components in the CQG method. Specifically, we investigate the effects of excluding certain elements: keeping only implicit negatives (no explicit negatives), without hard negatives, without easy negatives, and omitting the probing mechanism. The results of these experiments are presented in Table 12.

**Implicit Negatives.** To assess the impact of explicit negative samples, we modified the question generation prompt (Appendix A.1) to rely solely on implicit negatives. The modified prompt is shown as follows:

Table 12: Ablation study of different components in the CQG method on STS datasets.

| Model | Spearman Correlation ↑ (STS) | | | | | | | |
|---|---|---|---|---|---|---|---|---|
| | STS12 | STS13 | STS14 | STS15 | STS16 | STS-B | SICK-R | Avg. |
| **CQG-MBQA (Implicit Negative)** | 67.67 | 78.58 | 72.48 | 79.24 | 78.64 | 82.13 | 77.24 | 76.57 |
| **CQG-MBQA (w/o Hard Negative)** | 66.73 | 77.14 | 70.48 | 78.77 | 76.21 | 81.07 | 76.44 | 75.26 |
| **CQG-MBQA (w/o Easy Negative)** | 68.90 | 76.12 | 73.17 | 79.63 | 75.08 | 81.59 | 79.34 | 76.26 |
| **CQG-MBQA (w/o Probing)** | 68.29 | 77.92 | 71.17 | 79.80 | 77.06 | 81.33 | 76.52 | 76.01 |
| **CQG-MBQA** | 69.21 | 80.19 | 73.91 | 80.66 | 78.30 | 82.69 | 78.21 | 77.60 |

Table 13: Ablation study of different encoders (i.e., Stella (`stella_en_400M_v5`), GTE (`gte-large-en-v1.5`), and AnglE (`UAE-Large-V1`)) on STS datasets.

| Model | Spearman Correlation ↑ (STS) | | | | | | | |
|---|---|---|---|---|---|---|---|---|
| | STS12 | STS13 | STS14 | STS15 | STS16 | STS-B | SICK-R | Avg. |
| **CQG-MBQA (Stella)** | 54.45 | 75.66 | 64.92 | 76.13 | 74.20 | 74.01 | 73.37 | 70.39 |
| **CQG-MBQA (GTE)** | 63.34 | 73.28 | 68.24 | 78.45 | 73.64 | 75.08 | 73.20 | 72.18 |
| **CQG-MBQA (AnglE)** | 69.21 | 80.19 | 73.91 | 80.66 | 78.30 | 82.69 | 78.21 | 77.60 |

> Generate 10 simple yet insightful yes/no questions that determine the properties of an article, where for all questions, the answer will be "yes" for ALL the positive articles and "no" for general articles. Keep questions concise and avoid using complex sentence structures with "and" or "or" unless necessary.
> (*The rest of the prompt remains identical to the original version.*)

As shown in the first row of Table 12, the absence of explicit negative samples reduces performance from 77.60 to 76.57 (average across STS datasets). This demonstrates that explicit negatives are essential for refining the discriminative power of the generated questions.

**Without Hard Negatives.** In this experiment, we removed hard negative samples by setting both the hard negative samples per cluster ($n_h$) and the hard negative probe samples per question ($p_h$) to 0. The results, presented in the second row of Table 12, reveal a performance drop to 76.26 compared to 77.60 when both hard and easy negatives are included. This indicates that hard negatives, which are semantically similar to positives, play a crucial role in enhancing question discriminability.

**Without Easy Negatives.** To evaluate the significance of easy negatives, we excluded them by setting the easy negative samples per cluster ($n_e$) and the easy negative probe samples per question ($p_e$) to 0. As shown in the third row of Table 12, this exclusion results in a further decline in performance to 75.26, highlighting that easy negatives contribute to capturing broader semantic distinctions within the dataset.

**Without Probing.** Finally, we removed the probing mechanism for this experiment, instead relying solely on the original LLM-generated order of questions. The results, shown in the fourth row of Table 12, indicate a performance reduction to 76.01 compared to 77.60 when probing is included. This emphasizes the importance of the probing mechanism in ensuring high-quality, discriminative question selection.

### F.2 DIFFERENT ENCODERS

Beyond utilizing **AnglE** (`UAE-Large-V1`, 335M parameters) as the encoder in our **CQG-MBQA** framework, an advanced encoder that ranks among the top performers on the MTEB benchmark, we extended our evaluation to include two alternative encoders: **Stella** (`stella_en_400M_v5`[9], 435M parameters) and **GTE** (`gte-large-en-v1.5`, 434M parameters) (Zhang et al., 2024). These models, with parameter sizes comparable to **AnglE**, are also recognized for their strong performance on the MTEB benchmark, making them suitable candidates for comparison.

---

[9]`https://huggingface.co/dunzhang/stella_en_400M_v5`

Table 14: Ablation study of embedding quality for different percentages of best output dimensions to keep after filtering in the QAEmb method on STS datasets.

| Model | # Questions | Spearman Correlation ↑ (STS) | | | | | | | |
|---|---|---|---|---|---|---|---|---|---|
| | | STS12 | STS13 | STS14 | STS15 | STS16 | STS-B | SICK-R | Avg. |
| **QAEmb-MBQA (10%)** | 870 | 58.96 | 62.1 | 56.83 | 66.89 | 61.6 | 68.38 | 71.25 | 63.71 |
| **QAEmb-MBQA (20%)** | 1,938 | 59.41 | 62.74 | 57.05 | 68.37 | 62.53 | 69.81 | 71.6 | 64.5 |
| **QAEmb-MBQA (30%)** | 2,961 | 59.83 | 63.08 | 57.59 | 69.04 | 63.16 | 70.68 | 72.1 | 65.07 |
| **QAEmb-MBQA (40%)** | 4,042 | 59.77 | 63.32 | 57.66 | 69.17 | 63.08 | 70.97 | 72.21 | 65.17 |
| **QAEmb-MBQA (50%)** | 5,127 | 59.65 | 63.22 | 57.64 | 68.99 | 63.29 | 70.88 | 72.16 | 65.12 |
| **QAEmb-MBQA (60%)** | 6,064 | 59.58 | 63.22 | 57.69 | 68.88 | 63.0 | 71.03 | 72.12 | 65.07 |
| **QAEmb-MBQA (70%)** | 7,063 | 59.35 | 63.07 | 57.48 | 69.08 | 63.17 | 70.98 | 72.16 | 65.04 |
| **QAEmb-MBQA (80%)** | 8,103 | 59.49 | 63.2 | 57.67 | 69.15 | 63.1 | 71.05 | 72.24 | 65.13 |
| **QAEmb-MBQA (90%)** | 9,153 | 59.29 | 62.94 | 57.46 | 69.14 | 62.95 | 71.03 | 72.28 | 65.01 |
| **QAEmb-MBQA (Full)** | 10,654 | 59.40 | 63.19 | 57.68 | 69.29 | 63.18 | 71.33 | 72.33 | 65.20 |
| **CQG-MBQA (Full)** | 9,614 | 69.21 | 80.19 | 73.91 | 80.66 | 78.30 | 82.69 | 78.21 | 77.60 |

Table 15: Ablation study of cognitive load for different percentages of best output dimensions to keep after filtering in the QAEmb method on STS datasets.

| Model | # Questions | Cognitive Load ↓ (Normalized Cognitive Load ↓) (STS) | | | | | | | |
|---|---|---|---|---|---|---|---|---|---|
| | | STS12 | STS13 | STS14 | STS15 | STS16 | STS-B | SICK-R | Avg. |
| **QAEmb-MBQA (10%)** | 870 | 144 (16.6%) | 141 (16.2%) | 143 (16.4%) | 125 (14.4%) | 139 (16.0%) | 121 (13.9%) | 84 (9.7%) | 128 (14.7%) |
| **QAEmb-MBQA (20%)** | 1,938 | 310 (16.0%) | 303 (15.6%) | 309 (15.9%) | 272 (14.0%) | 299 (15.4%) | 263 (13.6%) | 184 (9.5%) | 277 (14.3%) |
| **QAEmb-MBQA (30%)** | 2,961 | 468 (15.8%) | 453 (15.3%) | 465 (15.7%) | 411 (13.9%) | 452 (15.3%) | 398 (13.4%) | 283 (9.6%) | 418 (14.1%) |
| **QAEmb-MBQA (40%)** | 4,042 | 633 (15.7%) | 612 (15.1%) | 631 (15.6%) | 557 (13.8%) | 609 (15.1%) | 541 (13.4%) | 387 (9.6%) | 567 (14.0%) |
| **QAEmb-MBQA (50%)** | 5,127 | 799 (15.6%) | 774 (15.1%) | 797 (15.5%) | 703 (13.7%) | 772 (15.1%) | 684 (13.3%) | 489 (9.5%) | 717 (14.0%) |
| **QAEmb-MBQA (60%)** | 6,064 | 935 (15.4%) | 907 (15.0%) | 933 (15.4%) | 828 (13.7%) | 908 (15.0%) | 803 (13.2%) | 575 (9.5%) | 841 (13.9%) |
| **QAEmb-MBQA (70%)** | 7,063 | 1,086 (15.4%) | 1,052 (14.9%) | 1,085 (15.4%) | 961 (13.6%) | 1,049 (14.9%) | 935 (13.2%) | 669 (9.5%) | 977 (13.8%) |
| **QAEmb-MBQA (80%)** | 8,103 | 1,241 (15.3%) | 1,202 (14.8%) | 1,239 (15.3%) | 1,097 (13.5%) | 1,199 (14.8%) | 1,067 (13.2%) | 766 (9.5%) | 1,116 (13.8%) |
| **QAEmb-MBQA (90%)** | 9,153 | 1,398 (15.3%) | 1,352 (14.8%) | 1,398 (15.3%) | 1,240 (13.5%) | 1,351 (14.8%) | 1,209 (13.2%) | 873 (9.5%) | 1,260 (13.8%) |
| **QAEmb-MBQA (Full)** | 10,654 | 1,626 (15.3%) | 1,571 (14.7%) | 1,625 (15.3%) | 1,443 (13.5%) | 1,577 (14.8%) | 1,408 (13.2%) | 1,018 (9.6%) | 1,467 (13.8%) |
| **CQG-MBQA (Full)** | 9,614 | 481 (5.0%) | 439 (4.6%) | 458 (4.8%) | 426 (4.4%) | 478 (5.0%) | 446 (4.6%) | 413 (4.3%) | 449 (4.7%) |

Table 13 presents a summary of the results for the STS task. While the alternative encoders deliver competitive performance, the **CQG-MBQA** model consistently achieves its best results when paired with **AnglE** as the encoder. This highlights **AnglE**'s effectiveness in capturing fine-grained semantic distinctions and its synergy with the **CQG-MBQA** framework.

# G  QUESTION FILTERING FOR QAEMB

In our work, we adapted the **QAEmb** method for general text embedding by incorporating LLM-based question generation and applying semantic deduplication as a post-processing step. In contrast, we developed a **Contrastive Question Generation (CQG)** method, which includes a probing mechanism to filter questions based on their discriminative ability and semantic deduplication as post-processing.

A notable difference between our **CQG** approach and the **QAEmb** baseline lies in the absence of an equivalent sparsity penalty in **QAEmb** for filtering question discriminability. This raises the question of whether the observed performance improvements in **CQG** stem primarily from the probing stage or from the generation of higher-quality questions. To disentangle these factors, we conducted two additional experiments: (1) Adding a Sparsity Penalty for **QAEmb** and (2) Removing the Probing Stage in **CQG**.

## G.1  ADDING A SPARSITY PENALTY FOR QAEMB

The **QAEmb** paper (Benara et al., 2024) proposes two post-hoc approaches for filtering questions, as described in the paragraph titled *"Post-hoc pruning of Q:"*

(1)  Using feature selection models (e.g., elastic net) with task labels.
(2)  Using an LLM to select questions relevant to a task description.

Since our work targets a general text embedding task without task-specific labels, we followed the second approach and developed an LLM-based method for filtering low-quality questions using a sparsity penalty approach. Specifically, we clustered the questions generated by **QAEmb** and employed an LLM to select subsets of questions from each cluster, varying the percentage of questions retained from 10% to 90% of the cluster size.

To ensure consistency and precision, we crafted the following detailed prompt, which includes task descriptions and explicit selection criteria:

> You are an expert in natural language processing and text embeddings. From the following list of questions, select the {num_to_keep} best questions that would be most effective for text embedding tasks.
>
> **Task Description:**
> Text embedding is a process where we convert text into numerical vectors that capture semantic meaning. Good questions for text embedding should help in:
> 1. Capturing the main topics and themes in texts
> 2. Understanding the semantic relationships between different pieces of text
> 3. Identifying key concepts and ideas
> 4. Distinguishing between different contexts and meanings
> 5. Enabling effective text similarity comparisons and search
>
> **Selection Criteria:**
> The selected questions should:
> 1. Be clear and well-formed
> 2. Cover diverse semantic aspects
> 3. Be general enough to apply to various texts
> 4. Avoid redundancy and similar phrasings
> 5. Focus on meaningful content rather than superficial details
> 6. Help in extracting semantic features useful for embedding generation
> 7. Exclude any questions that are unclear or ambiguous
>
> **Instructions:**
> - From the list below, select EXACTLY {num_to_keep} questions that best meet the above criteria.
> - Aim for diversity by choosing questions that cover a wide range of semantic features.
> - List only the numbers of the selected questions, separated by commas. For example: "1, 5, 8, 12".
> - Do not include any explanations or additional text in your response.
> - Your response should strictly follow the format specified.
>
> **Here are the questions:**
> {i}.{questions}

Using this sparsity penalty approach, we evaluated the **QAEmb-MBQA** model with different percentages of retained questions. As shown in Tables 14 and 15, the embedding quality remains comparable or slightly lower than **QAEmb-MBQA (Full)**, while the cognitive load is significantly reduced. This trade-off highlights the value of sparsity penalty in improving interpretability without severely impacting performance.

## G.2 REMOVING THE PROBING STAGE IN CQG

To isolate the effect of the probing mechanism, we conducted an experiment where the probing stage was removed from the **CQG** method. In this configuration, we used the original LLM-generated question order for evaluation. The results, labeled as **CQG-MBQA (w/o Probing)**, are presented in Table 12 in Appendix F.1.

When comparing **CQG-MBQA (w/o Probing)** with **CQG-MBQA** and **QAEmb-MBQA**, as reported in Table 1, we observe that **CQG-MBQA (w/o Probing)** still outperforms **QAEmb-MBQA**. However,

the probing mechanism contributes additional performance gains, underscoring its importance in refining the discriminative power of the questions.

