# OpenReview forum: "A General Framework for Producing Interpretable Semantic Text Embeddings"
_ICLR.cc/2025/Conference — ICLR 2025 Poster_

### Official Review · Reviewer_vj78 · 2024-10-18

**Soundness:** 3
**Presentation:** 3
**Contribution:** 3
**Rating:** 6
**Confidence:** 4

**Summary:**

The authors introduce CQG-MBQA, a general framework for producing interpretable semantic text embeddings. It builds these embeddings from a set of yes/no questions that are designed to be highly discriminative (by separating text clustered by a pre-trained embedding model). To improve efficiency, the answers to these yes/no questions are distilled into a smaller model. The CQG-MBQA model reveals improvements relative to baseline interpretable models.

**Strengths:**

- The authors study an interesting and important problem
- They obtain strong performance results with reasonable efficiency
- They evaluate interpretability and cognitive load well, in addition to more standard performance metrics

**Weaknesses:**

- The main issue seems to be the authors’ treatment of related work: the CGQ method generates questions through prompting and filters them based on their discriminative ability. The baseline QA-Emb also generates questions through prompting but filters them with a sparsity penalty. From their description, the authors don’t seem to implement the sparsity penalty, which likely skews the comparisons.
- The authors should discuss this [2023 style embeddings paper](https://arxiv.org/abs/2305.12696), which was an early precursor to the work here
- The authors should clarify whether distilling the yes/no answers into a single model is a novel contribution — the style embeddings paper & QA-Emb paper both seem to do this as well

**Questions:**

Can the authors provide a comparison with the baseline including the sparsity penalty? Maybe showing the performance as a function of the number of questions kept?

---

> ### Author Response · Authors · 2024-11-25
> **Response to Reviewer vj78 [1/3]**
>
> We sincerely thank the reviewer for their insightful feedback and valuable suggestions.
> Based on these recommendations, we have made several revisions and conducted additional experiments.
> Below is a detailed response outlining the changes and improvements we have implemented.
>
> ---
>
> > The authors should discuss this 2023 style embeddings paper, which was an early precursor to the work here.
>
> Thank you for bringing this paper to our attention. It employs the generative and instruction-following capabilities of LLMs to analyze and summarize text styles, using LLM outputs to train a smaller language model for enhanced efficiency. Additionally, post-processing techniques are applied to identify the most effective style attributes.
>
> This work is indeed highly relevant and pioneering for our research. We will ensure it is appropriately cited and thoroughly discussed in the next revision, which we plan to complete by the end of November 27.
>
> ---
>
> > The authors should clarify whether distilling the yes/no answers into a single model is a novel contribution — the style embeddings paper & QA-Emb paper both seem to do this as well.
>
> Thank you for your feedback. We would like to clarify that distilling LLM-generated answers into a single model is a practice employed in both the style embeddings paper and the QAEmb paper. Rather than positioning this as a novel contribution, our goal is to highlight its role in creating a practical and complete pipeline.
>
> Specifically, while the QAEmb paper conducted distillation experiments for the fMRI task, it did not extend this approach to other tasks.
> Our work focuses on developing a general and practical framework for text embeddings, prioritizing cost-effectiveness and scalability.
> We will ensure this point is clearly emphasized in the revised version to prevent any potential misleading.
>
> ---
>
> > The main issue seems to be the authors’ treatment of related work: the CGQ method generates questions through prompting and filters them based on their discriminative ability. The baseline QA-Emb also generates questions through prompting but filters them with a sparsity penalty. From their description, the authors don’t seem to implement the sparsity penalty, which likely skews the comparisons.
> > Can the authors provide a comparison with the baseline including the sparsity penalty? Maybe showing the performance as a function of the number of questions kept?
>
> Thank you for raising this concern.
> The original QAEmb paper focuses on task-specific text embeddings with task labels, whereas our framework aims to create general text embeddings, similar to pre-trained text encoders.
>
> In our work, we adapted the QAEmb method for general text embedding by incorporating LLM-based question generation and applying semantic deduplication as post-processing.
> In contrast, we developed a contrastive question generation (CQG) method, which includes a probing mechanism (to filter questions based on their discriminative ability) and semantic deduplication as post-processing.
>
> We acknowledge that the absence of a sparsity penalty equivalent in the QAEmb baseline for filtering question discriminative ability might leave uncertainty about whether the observed improvements stem from the probing stage or higher-quality question generation. To address this, we conducted two additional sets of experiments:
>
> 1. **Adding Sparsity Penalty for QAEmb**
>
> The paragraph of "Post-hoc pruning of $Q$" in the original QAEmb paper proposes two approaches for filtering:
>
> - (1) Using feature selection models (e.g., elastic net) with task labels.
> - (2) Using an LLM to select questions relevant to a task description.
>
> Since our work focuses on a general text embedding task without task labels, we followed approach (2) and developed a LLM-based method to filter low-quality questions using a sparsity penalty approach.
> Specifically, we clustered the questions generated by QAEmb and used the LLM to select subsets of questions within each cluster, varying the percentage of questions retained (from 10% to 90% of the cluster size).

---

> ### Author Response · Authors · 2024-11-25
> **Response to Reviewer vj78 [2/3]**
>
> To ensure consistency, we carefully designed the following prompt, which includes a detailed task description and selection criteria:
>
> ```
> You are an expert in natural language processing and text embeddings. From the following list of questions, select the **{num_to_keep} best questions** that would be most effective for text embedding tasks.
>
> **Task Description:**
>
> Text embedding is a process where we convert text into numerical vectors that capture semantic meaning. Good questions for text embedding should help in:
>
> 1. **Capturing the main topics and themes in texts**
> 2. **Understanding the semantic relationships between different pieces of text**
> 3. **Identifying key concepts and ideas**
> 4. **Distinguishing between different contexts and meanings**
> 5. **Enabling effective text similarity comparisons and search**
>
> **Selection Criteria:**
>
> The selected questions should:
>
> 1. **Be clear and well-formed**
> 2. **Cover diverse semantic aspects**
> 3. **Be general enough to apply to various texts**
> 4. **Avoid redundancy and similar phrasings**
> 5. **Focus on meaningful content rather than superficial details**
> 6. **Help in extracting semantic features useful for embedding generation**
> 7. **Exclude any questions that are unclear or ambiguous**
>
> **Instructions:**
>
> - From the list below, select **EXACTLY {num_to_keep} questions** that best meet the above criteria.
> - **Aim for diversity** by choosing questions that cover a wide range of semantic features.
> - **List only the numbers** of the **selected questions**, separated by commas. For example: "1, 5, 8, 12".
> - **Do not include any explanations or additional text** in your response.
> - **Your response should strictly follow the format specified.**
>
> **Here are the questions:**
>
> {questions}
> ```
>
> Using this sparsity penalty approach, we evaluated the QAEmb-MBQA model with different percentages of # questions as output dimensions.
>
> The results are presented in **Tables 1 and 2** below. We observe that while the embedding quality remains comparable or slightly lower than QAEmb-MBQA (Full), the cognitive load is notably reduced.
>
> **Table 1: Spearman Correlation vs. the Percentage of # Questions Retained**
>
> | Model | # Questions | STS12 | STS13 | STS14 | STS15 | STS16 | STS-B | SICK-R | Avg. |
> | ----- | ----------- | ----- | ----- | ----- | ----- | ----- | ----- | ------ | ---- |
> |QAEmb-MBQA (10%)|870|58.96|62.1|56.83|66.89|61.6|68.38|71.25|63.71|
> |QAEmb-MBQA (20%)|1938|59.41|62.74|57.05|68.37|62.53|69.81|71.6|64.5|
> |QAEmb-MBQA (30%)|2961|59.83|63.08|57.59|69.04|63.16|70.68|72.1|65.07|
> |QAEmb-MBQA (40%)|4042|59.77|63.32|57.66|69.17|63.08|70.97|72.21|65.17|
> |QAEmb-MBQA (50%)|5127|59.65|63.22|57.64|68.99|63.29|70.88|72.16|65.12|
> |QAEmb-MBQA (60%)|6064|59.58|63.22|57.69|68.88|63.0|71.03|72.12|65.07|
> |QAEmb-MBQA (70%)|7063|59.35|63.07|57.48|69.08|63.17|70.98|72.16|65.04|
> |QAEmb-MBQA (80%)|8103|59.49|63.2|57.67|69.15|63.1|71.05|72.24|65.13|
> |QAEmb-MBQA (90%)|9153|59.29|62.94|57.46|69.14|62.95|71.03|72.28|65.01|
> |QAEmb-MBQA (Full)|10654 |59.40|63.19|57.68|69.29|63.18|71.33|72.33|65.20|
> | CQG-MBQA (Full) |9614  | 69.21 | 80.19 | 73.91 | 80.66 | 78.30 | 82.69 | 78.21  | 77.60 |
>
> **Table 2: Cognitive Load vs. the Percentage of # Questions Retained**
>
> | Model | # Questions | STS12 | STS13 | STS14 | STS15 | STS16 | STS-B | SICK-R | Avg. |
> | ----- | ----------- | ----- | ----- | ----- | ----- | ----- | ----- | ------ | ---- |
> |QAEmb-MBQA (10%)|870|144|141|143|125|139|121|84|128|
> |QAEmb-MBQA (20%)|1938|310|303|309|272|299|263|184|277|
> |QAEmb-MBQA (30%)|2961|468|453|465|411|452|398|283|418|
> |QAEmb-MBQA (40%)|4042|633|612|631|557|609|541|387|567|
> |QAEmb-MBQA (50%)|5127|799|774|797|703|772|684|489|717|
> |QAEmb-MBQA (60%)|6064|935|907|933|828|908|803|575|841|
> |QAEmb-MBQA (70%)|7063|1086|1052|1085|961|1049|935|669|977|
> |QAEmb-MBQA (80%)|8103|1241|1202|1239|1097|1199|1067|766|1116|
> |QAEmb-MBQA (90%)|9153|1398|1352|1398|1240|1351|1209|873|1260|
> |QAEmb-MBQA (Full)|10654 |1626|1571|1625|1443|1577|1408|1018|1467|
> | CQG-MBQA (Full) |9614  | 481 | 439 | 458 | 426 | 478 | 446 | 413  | 449 |

---

> ### Author Response · Authors · 2024-11-25
> **Response to Reviewer vj78 [3/3]**
>
> 2. **Removing Probing Stage for CQG**
>
> To isolate the impact of the probing mechanism, we removed it from the CQG method and evaluated performance using the original LLM-generated question order. The results are presented in **Tables 3 and 4** below.
>
> From this comparison, we observe that CQG-MBQA without probing still outperforms QAEmb-MBQA, though the probing mechanism provides additional performance improvements.
>
> In our next revision (to be submitted by the end of November 27), we will:
> - Clearly describe how the QAEmb baseline was adapted for general text embedding.
> - Highlight the fundamental differences in task settings between QAEmb and CQG.
> - Include the results of these additional experiments to ensure transparency and fairness in the comparisons in the appendix.
>
> **Table 3: Spearman Correlation with/without Probing**
>
> | Model                 | STS12 | STS13 | STS14 | STS15 | STS16 | STS-B | SICK-R | Avg.  |
> | --------------------- | ----- | ----- | ----- | ----- | ----- | ----- | ------ | ----- |
> | QAEmb-MBQA            | 59.40 | 63.19 | 57.68 | 69.29 | 63.18 | 71.33 | 72.33  | 65.20 |
> | CQG-MBQA (No Probing) | 68.29 | 77.92 | 71.17 | 79.80 | 77.06 | 81.33 | 76.52  | 76.01 |
> | CQG-MBQA              | 69.21 | 80.19 | 73.91 | 80.66 | 78.30 | 82.69 | 78.21  | 77.60 |
>
> **Table 4: Cognitive Load with/without Probing**
>
> | Model                 | STS12 | STS13 | STS14 | STS15 | STS16 | STS-B | SICK-R | Avg. |
> | --------------------- | ----- | ----- | ----- | ----- | ----- | ----- | ------ | ---- |
> | QAEmb-MBQA            | 1626  | 1571  | 1625  | 1443  | 1577  | 1408  | 1018   | 1467 |
> | CQG-MBQA (No Probing) | 194   | 181   | 187   | 174   | 199   | 183   | 167    | 184  |
> | CQG-MBQA              | 481   | 439   | 458   | 426   | 478   | 446   | 413    | 449  |
>
> ---
>
> We hope our responses and additional experiments address your concerns and clarify our contributions. Thank you again for your thoughtful feedback, which has greatly helped improve our work. We look forward to any further suggestions you may have.

---

> > ### Comment · Reviewer_vj78 · 2024-11-26
> >
> > I thank the authors for their response & added comparison with QA-Emb when filtering questions. I have increased my score (from 5 to 6).

---

> > > ### Author Response · Authors · 2024-11-28
> > >
> > > Thank you for taking the time to review our response and for increasing your score. We greatly appreciate your recognition of the additional comparison with QA-Emb and your constructive feedback, which has been invaluable in improving our work. Please do not hesitate to reach out if you have any further questions or suggestions.

---

### Official Review · Reviewer_8TCb · 2024-10-31

**Soundness:** 2
**Presentation:** 3
**Contribution:** 2
**Rating:** 5
**Confidence:** 4

**Summary:**

This work proposes a "framework" for creating interpretable semantic embeddings. They tackle the important and relevant problem of creating embeddings that are useful for search & clustering but also understandable to humans.

**Strengths:**

- This paper tackles the important problem of creating interpretable text embeddings
- Some of the steps laid out in the "framework" explanation will be useful for other practitioners
- The source code is available and could be used by other researchers and engineers to build systems for interpretable embeddings
- Consideration of the tradeoff between interpretability and quality is interesting – although I have qualms with the "cognitive load" measurement of interpretability, which are mentioned below.

**Weaknesses:**

- I find it important to point out that this paper isn't really proposing a "framework" in a very general sense; it's much closer to a method (and in fact the authors interchange the two words liberally throughout the paper). For this reason I object to calling it a framework at all and would prefer the paper to be about CQG-MBQA, which is an interesting and apparently effective method for interpretable
 text embeddings.
- As a related point, the organization is confusing. Currently the paper mixes together the "framework" part (which should be a general process for producing interpretable embeddings) with the "method" part (about CQG-MBQA) and also some "experimental setup" and "results"  (all in Section 3). As one example of the murky sectional boundaries, is post-processing really a necessary step of the framework?
- I'm also not sure if the Case Study is exactly a Case Study.
- The cognitive load metric seems unprincipled and lacks grounding in real cognitive science.
- Cognitive load is simply the number of overlapping "yes" answers (or 1s in the embeddings) between the representations of a pair from an STS dataset. It is highly dependent on dimensionality and sparsity (Figure 4 & 5). It also doesn't really make sense because the interpretability of an embedding should depend on how many yes's there are for a pair *compared to other pairs*; embeddings cannot be understood simply by looking at the inner product of a pair of embeddings.
- Many of the important design decisions in the framework are not ablated. Is filtering important? How much does choosing positive and negative samples matter, or clustering? How much does training a surrogate model affect performance?
- This is not necessary to me for accepting the paper, but a real human study could be crucial for arguing that these embeddings are in fact more interpretable
- Due to the complicated system and lack of ablations, it is not easy to understand why these embeddings outperform other interpretable embeddings such as QAEmb
- Unclear cost analysis of running this method on a downstream dataset

I think this citation could be relevant:
- Learning Interpretable Style Embeddings via Prompting LLMs (Patel et al., EMNLP Findings 2023)

**Questions:**

- What inspired this measurement of cognitive load?
- How much inference time does it take to run on the retrieval datasets?
- How were the retrieval datasets chosen?
- How much does the QA model quality affect embedding quality?

---

> ### Author Response · Authors · 2024-11-25
> **Response to Reviewer 8TCb [1/4]**
>
> We sincerely thank the reviewer for their thoughtful and constructive feedback. Your comments have provided valuable insights that have helped us refine our work and identify areas for improvement.
> In response, we have conducted additional experiments, addressed specific concerns, and provided detailed explanations to clarify our design decisions and their impact.
> Below, we outline our responses and highlight the changes made based on your feedback.
>
> ---
>
> > Cognitive load is simply the number of overlapping "yes" answers (or 1s in the embeddings) between the representations of a pair from an STS dataset. It is highly dependent on dimensionality and sparsity (Figure 4 & 5). It also doesn't really make sense because the interpretability of an embedding should depend on how many yes's there are for a pair compared to other pairs; embeddings cannot be understood simply by looking at the inner product of a pair of embeddings.
>
> Thank you for your valuable suggestion. We have provided **Table 1** below, which shows the normalized cognitive load (in percentage) for the STS task. These normalized results indicate that our CQG-MBQA framework consistently achieves a lower cognitive load compared to the QAEmb-MBQA method across different datasets. Furthermore, the observed trend aligns closely with the results presented in Table 5 of our original paper.
>
> In the revised version of the paper (to be submitted by the end of November 27), we will update Table 5 to include this additional information.
>
> **Table 1: The Normalized Cognitive Load (in percentage) for the STS task**
> | Model         |   STS12 |   STS13 |   STS14 |   STS15 |   STS16 |   STS-B |   SICK-R |   Avg. |
> |:--------------|--------:|--------:|--------:|--------:|--------:|--------:|---------:|-------:|
> | Bag-of-Tokens |    0.03 |    0.01 |    0.02 |    0.02 |    0.03 |    0.02 |     0.02 |   0.02 |
> | QAEmb-MBQA    |   15.26 |   14.75 |   15.25 |   13.54 |   14.80 |   13.22 |     9.56 |  13.77 |
> | CQG-MBQA      |    5.00 |    4.57 |    4.76 |    4.43 |    4.97 |    4.64 |     4.30 |   4.67 |
>
> ---
>
> > Many of the important design decisions in the framework are not ablated. Is filtering important? How much does choosing positive and negative samples matter, or clustering? How much does training a surrogate model affect performance?
> > Due to the complicated system and lack of ablations, it is not easy to understand why these embeddings outperform other interpretable embeddings such as QAEmb
>
> We conducted extensive ablation studies to address these questions:
>
> 1. **Comparison of CQG with and without the probing mechanism**
>
> In our CQG-MBQA framework, we employ the probing mechanism for filtering the low-quality questions.
> To study the effectiveness of filtering, we conducted ablation studies to evaluate the effect of the probing mechanism.
>
> For this experiment, we removed the probing mechanism and used the original LLM-generated order.
> Results in **Table 2** below indicate a performance drop to **76.01**, compared to **77.60** when the probing mechanism is included.
>
> **Table 2: Spearman Correlation on STS Datasets with/without the Probing Mechanism**
>
> | Model             | STS12 | STS13 | STS14 | STS15 | STS16 | STS-B | SICK-R | Avg.  |
> | ----------------- | ----- | ----- | ----- | ----- | ----- | ----- | ------ | ----- |
> | CQG-MBQA without Probing  | 68.29 | 77.92 | 71.17 | 79.80 | 77.06 | 81.33 | 76.52  | 76.01 |
> | CQG-MBQA with Probing     | 69.21 | 80.19 | 73.91 | 80.66 | 78.30 | 82.69 | 78.21  | 77.60 |

---

> ### Author Response · Authors · 2024-11-25
> **Response to Reviewer 8TCb [2/4]**
>
> 2. **Comparison between vanilla CQG with different types of samples:**
>
> We conducted ablation studies to evaluate the impact of different types of samples on the CQG method's performance.
> Specifically, we tested three scenarios, with results summarized in **Table 3** below:
>
> - **(a) Only positive samples:**
>   - We modified the question generation prompt to exclude explicit negative samples:
>     - **Original Prompt:**
>       `...where for all questions, the answer will be "yes" for ALL the positive articles and "no" for ALL the negative articles.`
>     - **Modified Prompt:**
>       `...where for all questions, the answer will be "yes" for ALL the positive articles and "no" for general articles.`
>   - Results show that explicit negative samples improve performance from **76.57** to **77.60** (average across STS datasets).
>
> - **(b) No hard negatives:**
>   - We set the hard negative samples per cluster ($n_h$) and the hard negative probe samples per question ($p_h$) to **0**.
>   - Results show performance drops to **76.26**, compared to **77.60** with both hard and easy negatives included.
>
> - **(c) No easy negatives:**
>   - We set the easy negative samples per cluster ($n_e$) and the easy negative probe samples per question ($p_e$) to **0**.
>   - Results show performance drops to **75.26**, compared to **77.60** with both hard and easy negatives included.
>
> From **Table 3**, it is evident that the full CQG-MBQA method (including positive samples, hard negatives, and easy negatives) yields the highest question quality and achieves the best downstream performance on the STS datasets.
>
> **Table 3: Spearman Correlation on STS Datasets for Different Types of Samples**
> | Model             | STS12 | STS13 | STS14 | STS15 | STS16 | STS-B | SICK-R | Avg.  |
> | ----------------- | ----- | ----- | ----- | ----- | ----- | ----- | ------ | ----- |
> | Implicit Negative | 67.67 | 78.58 | 72.48 | 79.24 | 78.64 | 82.13 | 77.24  | 76.57 |
> | No Hard Negative  | 66.73 | 77.14 | 70.48 | 78.77 | 76.21 | 81.07 | 76.44  | 75.26 |
> | No Easy Negative  | 68.90 | 76.12 | 73.17 | 79.63 | 75.08 | 81.59 | 79.34  | 76.26 |
> | CQG-MBQA (Full)   | 69.21 | 80.19 | 73.91 | 80.66 | 78.30 | 82.69 | 78.21  | 77.60 |
>
> 3. **Comparison between the MBQA surrogate model and directly using the LLM’s outputs**
>
> Unfortunately, conducting experiments using LLM output on the STS datasets poses significant computational and financial challenges.
> Specifically, it would require approximately **47.7 million** API calls, even if we group 20 questions into a single prompt. This would entail thousands of dollars in API credits (assuming 300 tokens per API call) or at least several months of computation time using a local LLM (assuming each API call takes 0.1 seconds).
>
> As an alternative, we have evaluated the accuracy of our MBQA model in replicating LLM outputs.
> In **Table 6** of the appendix in our original paper, we demonstrate that the classification accuracy is **96%**, indicating that the MBQA model is sufficiently accurate when compared to the LLM output.
>
> In addition to the STS, retrieval, and clustering tasks, we have also tested our framework on four additional downstream tasks from the MTEB benchmark: classification, pair classification, reranking, and summarization evaluation. The results are provided in **Tables 4-7** below.
>
> As shown in **Tables 4-7**, our framework consistently outperforms existing interpretable text embedding models and achieves performance comparable to many advanced black-box models across all tested downstream tasks.
>
> The strong performance of our model on seven downstream tasks (three reported in our original paper and these four additional tasks) provides indirect evidence of the MBQA model's competitiveness and reliability. We hope this addresses your concern and clarifies the robustness of our approach.

---

> ### Author Response · Authors · 2024-11-25
> **Response to Reviewer 8TCb [3/4]**
>
> **Table 4: the Classification Task with 12 Datasets**
>
> | Model     | AmazonCounterfactualClassification | AmazonPolarityClassification | AmazonReviewsClassification | Banking77Classification |
> | --------------- | ---------------------------------- | ---------------------------- | --------------------------- | ----------------------- |
> | BERT      | 74.25  | 71.33    | 33.56   | 63.41     |
> | SimCSE (Unsup.) | 67.09  | 74.48    | 33.85   | 73.55     |
> | GloVe     | 56.91  | 60.32    | 29.67   | 67.69     |
> | SimCSE (Sup.)   | 75.75  | 82.47    | 39.6    | 75.76     |
> | SBERT (New)     | 65.28  | 62.98    | 30.79   | 80.4      |
> | AnglE     | 75.55  | 92.84    | 48.29   | 87.69     |
> | OpenAI    | 75.94  | 86.72    | 44.78   | 80.66     |
> | BoT       | 78.87  | 55.28    | 27.95   | 60.63     |
> | QAEmb-MBQA      | 59.81  | 84.43    | 40.31   | 77.72     |
> | CQG-MBQA  | 62.62  | 93.66    | 45.39   | 83.45     |
>
> | Model     | EmotionClassification | ImdbClassification | MassiveIntentClassification | MassiveScenarioClassification |
> |-----------------|-----------------------|--------------------|-----------------------------|-------------------------------|
> | BERT      | 35.28   | 65.35  | 59.88   | 64.28     |
> | SimCSE (Unsup.) | 42.22   | 69.63  | 59.84   | 66.25     |
> | GloVe     | 36.93   | 62.57  | 56.19   | 66.03     |
> | SimCSE (Sup.)   | 44.81   | 73.53  | 65.95   | 70.78     |
> | SBERT (New)     | 41.17   | 59.76  | 67.15   | 74.58     |
> | AnglE     | 51.75   | 92.78  | 76.5    | 79.75     |
> | OpenAI    | 48.74   | 77.98  | 70.15   | 75.33     |
> | BoT       | 22.17   | 53.32  | 48.79   | 49.63     |
> | QAEmb-MBQA      | 39.68   | 89.27  | 62.52   | 68.87     |
> | CQG-MBQA  | 46.04   | 92.8 | 70.2    | 74.9      |
>
>
> | Model     | MTOPDomainClassification | MTOPIntentClassification | ToxicConversationsClassification | TweetSentimentExtractionClassification | Avg.  |
> | --------------- | ------------------------ | ------------------------ | -------------------------------- | -------------------------------------- | ----- |
> | BERT      | 82.63      | 68.14      | 70   | 51.81      | 61.66 |
> | SimCSE (Unsup.) | 81.71      | 59.23      | 68.82  | 53.36      | 62.50 |
> | GloVe     | 79.11      | 55.85      | 65.4 | 50.8 | 57.29 |
> | SimCSE (Sup.)   | 84.29      | 63.14      | 72.04  | 59.73      | 67.32 |
> | SBERT (New)     | 91.9 | 62.84      | 67.47  | 54.25      | 63.21 |
> | AnglE     | 94.02      | 76.92      | 71.09  | 59.75      | 75.58 |
> | OpenAI    | 92.13      | 64.68      | 72.29  | 61.81      | 70.93 |
> | BoT       | 72.77      | 58.41      | 53.24  | 43.59      | 52.05 |
> | QAEmb-MBQA      | 80.95      | 60.23      | 59.91  | 56.03      | 64.98 |
> | CQG-MBQA  | 89.79      | 66.95      | 60.79  | 61.48      | 70.67 |
>
>
> **Table 5: the Pair Classification Task with 3 Datasets**
>
> | Model     | SprintDuplicateQuestions | TwitterSemEval2015 | TwitterURLCorpus | Avg.  |
> | --------------- | ------------------------ | ------------------ | ---------------- | ----- |
> | BERT      | 36.81      | 55.9 | 76.29      | 56.33 |
> | SimCSE (Unsup.) | 78.03      | 61.01  | 81.37      | 73.47 |
> | GloVe     | 86.96      | 53.12  | 77.35      | 72.48 |
> | SimCSE (Sup.)   | 73.04      | 67.75  | 83.89      | 74.89 |
> | SBERT (New)     | 92.58      | 70.02  | 84.77      | 82.46 |
> | AnglE     | 97.24      | 78.17  | 86.33      | 87.25 |
> | OpenAI    | 92.17      | 75.28  | 87.22      | 84.89 |
> | BoT       | 83.33      | 59.82  | 78.63      | 73.26 |
> | QAEmb-MBQA      | 43.71      | 60.04  | 73.21      | 59.65 |
> | CQG-MBQA  | 81.77      | 67.42  | 79.13      | 76.11 |
> **Table 6: the Reranking Task with 3 Datasets**
>
> | Model     | AskUbuntuDupQuestions | MindSmallReranking | SciDocsRR | StackOverflowDupQuestions | Avg.  |
> | --------------- | --------------------- | ------------------ | --------- | ------------------------- | ----- |
> | BERT      | 45.84   | 28.37  | 64.94     | 34.62 | 43.44 |
> | SimCSE (Unsup.) | 51.57   | 28.62  | 66.33     | 39.35 | 46.47 |
> | GloVe     | 49.57   | 27.01  | 62.56     | 34.03 | 43.29 |
> | SimCSE (Sup.)   | 51.8    | 29.3 | 70.14     | 38.9  | 47.54 |
> | SBERT (New)     | 64.06   | 31.02  | 87.2      | 51.47 | 58.44 |
> | AnglE     | 64.2    | 32.51  | 87.49     | 55.32 | 59.88 |
> | OpenAI    | 62.05   | 31.45  | 81.22     | 50.54 | 56.32 |
> | BoT       | 49.28   | 23.99  | 56.20     | 37.99 | 41.86 |
> | QAEmb-MBQA      | 54.70   | 28.73  | 70.86     | 40.81 | 48.78 |
> | CQG-MBQA  | 59.61   | 30.83  | 81.72     | 47.33 | 54.87 |
>
> **Table 7: the Summarization Task with 1 Dataset**
>
> | Model     | SummEval |
> | --------------- | -------- |
> | BERT      | 29.82    |
> | SimCSE (Unsup.) | 31.15    |
> | GloVe     | 28.87    |
> | SimCSE (Sup.)   | 31.17    |
> | SBERT (New)     | 27.9     |
> | AnglE     | 32.03    |
> | OpenAI    | 30.8     |
> | BoT       | 28.2     |
> | QAEmb-MBQA      | 28.57    |
> | CQG-MBQA  | 30.41    |

---

> ### Author Response · Authors · 2024-11-25
> **Response to Reviewer 8TCb [4/4]**
>
> ---
>
> > Unclear cost analysis of running this method on a downstream dataset
> > How much inference time does it take to run on the retrieval datasets?
>
> Thank you for your suggestion. We measured inference time (in seconds) on 9 MTEB benchmark retrieval and clustering datasets using a single H100 GPU. The results are presented in **Table 8** below.
>
> Our model is highly efficient as the MBQA framework requires only a single Transformer forward pass, with minimal additional overhead from small classification heads.
>
> **Table 8: Inference time (in seconds) on 9 MTEB benchmark retrieval and clustering datasets**
>
> | Datasets    | ArguAna | FiQA-2018 | NFCorpus | SCIDOCS | SciFact | TwentyNewsgroupsClustering | StackExchangeClusteringP2P | BiorxivClusteringS2S | RedditClustering |
> | -------- | ------- | --------- | -------- | ------- | ------- | -------------------------- | -------------------------- | -------------------- | ---------------- |
> | CQG-MBQA | 133     | 1061      | 112      | 446     | 126     | 754                        | 1493                       | 724                  | 5452             |
>
> ---
>
> > I think this citation could be relevant:
> > Learning Interpretable Style Embeddings via Prompting LLMs (Patel et al., EMNLP Findings 2023)
>
> Thank you for bringing this paper to our attention. It employs the generative and instruction-following capabilities of LLMs to analyze and summarize text styles, using LLM outputs to train a smaller language model for enhanced efficiency. Additionally, post-processing techniques are applied to identify the most effective style attributes.
>
> This work is indeed highly relevant and pioneering for our research. We will ensure it is appropriately cited and thoroughly discussed in the next revision, which we plan to complete by the end of November 27.
>
> ---
>
> > What inspired this measurement of cognitive load?
>
> Thank you for your question. The cognitive load measurement in our study is inspired by the approach introduced in the COGAM paper [1], which uses the number of visual chunks as a proxy for cognitive load in the context of model explanations.
>
> Similarly, in our work, we use the number of questions a user needs to read as a proxy for cognitive load. This approach aligns with the idea of measuring the cognitive effort required to interpret or interact with the system.
>
> [1] COGAM: Measuring and Moderating Cognitive Load in Machine Learning Model Explanations (CHI 2020)
>
> ---
>
> > How were the retrieval datasets chosen?
>
> We conducted experiments on retrieval datasets from the MTEB benchmark, which includes datasets from the BEIR benchmark. To ensure a robust evaluation, we selected a diverse range of datasets of reasonable size from MTEB.
>
> Additionally, to further enhance diversity, we incorporated an extra dataset for news retrieval, as the news datasets included in BEIR are private and not publicly accessible.
>
> ---
>
> > How much does the QA model quality affect embedding quality?
>
> Thank you for raising this point. The backbone encoder of the MBQA model plays a crucial role in determining the quality of the QA model. To explore this further, in addition to `UAE-Large-V1`, we evaluated two alternative encoders: `stella_en_400M_v5` and `gte-large-en-v1.5`. Both of these models are highly ranked on the MTEB benchmark and have a comparable parameter size (approximately 400M–500M).
>
> A summary of the results on the STS task is provided in **Table 9** below.
> The results indicate that while these alternative encoders perform reasonably well, `UAE-Large-V1` consistently achieves the best performance.
>
> We acknowledge that the impact of different encoders remains an important area for further exploration. However, due to time constraints, we focused on using a recently published, high-performing model (`UAE-Large-V1`) for our experiments to ensure robust and timely evaluations.
>
> **Table 9: Spearman Correlation on STS Datasets for Different Encoder Models**
> | Model                        | STS12 | STS13 | STS14 | STS15 | STS16 | STS-B | SICK-R | Avg.  |
> | ---------------------------- | ----- | ----- | ----- | ----- | ----- | ----- | ------ | ----- |
> | CQG-MBQA (stella_en_400M_v5) | 54.45 | 75.66 | 64.92 | 76.13 | 74.20 | 74.01 | 73.37  | 70.39 |
> | CQG-MBQA (gte-large-en-v1.5) | 63.34 | 73.28 | 68.24 | 78.45 | 73.64 | 75.08 | 73.20  | 72.18 |
> | CQG-MBQA (UAE-Large-V1)      | 69.21 | 80.19 | 73.91 | 80.66 | 78.30 | 82.69 | 78.21  | 77.60 |
>
> ---
>
> Thank you once again for your thoughtful and constructive feedback. Your insights have been invaluable in helping us refine our work and address key areas of improvement. We look forward to any further suggestions or comments you may have.

---

> > ### Comment · Reviewer_8TCb · 2024-11-26
> >
> > Hi, thanks for the response; I read through it all this morning. I think my most important concerns remain unanswered (such as issues with the cognitive load metric) so I elect to keep my score where it is. Thanks for putting so much effort into this rebuttal process nonetheless.

---

> > > ### Author Response · Authors · 2024-11-28
> > >
> > > Thank you for taking the time to review our responses and for providing thoughtful feedback throughout the process. We understand and respect your decision to maintain your score, and your feedback has been invaluable in guiding our revisions and highlighting areas for further improvement. Please do not hesitate to reach out if any further questions or concerns arise.

---

### Official Review · Reviewer_RC4T · 2024-11-03

**Soundness:** 3
**Presentation:** 3
**Contribution:** 3
**Rating:** 6
**Confidence:** 3

**Summary:**

The paper introduces CQG-MBQA, a framework designed to produce interpretable semantic text embeddings for diverse NLP tasks. The framework uses Contrastive Question Generation (CQG) to automatically generate meaningful yes/no questions without domain experts. The Multi-task Binary Question Answering (MBQA) model answers these questions, producing embeddings with human-interpretable dimensions at a much lower cost than answering with LLMs, while maintaining comparable accuracy. The authors validate CQG-MBQA through experiments, comparing it to black-box and interpretable models across STS, retrieval, and clustering. The experimental results show that CQG-MBQA offers better embedding quality than existing interpretable models and provides comparable embedding quality to several black-box models, maintaining high interpretability and efficiency.

**Strengths:**

The ideas behind CQG and MBQA are novel and effective, supported by thoughtful experiments and ablation studies. The paper is clearly written and well-structured. Given the increasing demand for model transparency, CQG-MBQA could have significant implications and represent a meaningful approach that would be of interest to the ICLR audience.

- The paper builds upon QAEmb with important innovations: a contrastive approach to question generation that improves discrimination using positive, hard negative, and easy negative samples for fine-grained specificity, and a multi-task model for efficient inference without the use of LLMs.
- The technical quality is demonstrated through comprehensive empirical evaluation across multiple tasks and strong baselines including both interpretable and black-box models, with clear cost analysis showing significant efficiency gains through MBQA during inference. For reproducibility, detailed implementation specifics and code are provided.

**Weaknesses:**

- In retrieval tasks, there is a significant performance gap compared to black-box models, and the performance is also lower than BM25. Therefore, additional performance comparisons are needed when applying them to various downstream tasks such as sentiment classification and retrieval.
- Lack of ablation studies to assess the efficacy of the proposed approach
  - lack of comparison between different models in Figure 4 and 5, and lack of comparison between the MBQA method and directly using the LLM’s outputs.
  - comparison between vanilla CQG with positive and hard/easy negative, and CQG with positive and negative samples
  - comparison between having and not having the probing mechanism to refine the generated questions
- Also, because the cognitive load is defined using the dot product, this measure would be directly influenced by the total number of questions. A normalized version (e.g., dot product divided by number of questions) would provide a fairer comparison across different interpretable models in Table 5.
- Having cost analysis would be beneficial as MBQA requires significant LLM inferences (or API calls) during training time and even more may be required during Post-processing (probing).
- Including a case study on bad examples would also be beneficial—for instance, displaying cases where two texts result in similar embeddings even when those two texts do not necessarily have similar semantics. Are they completely off? Or how could one improve your approach?

**Questions:**

- Did you evaluate your model on the fMRI task presented in the QAEmb paper for a direct performance comparison?
- How do variations in the number of initial clusters (k) or the use of different clustering methods affect the performance of CQG?

---

> ### Author Response · Authors · 2024-11-25
> **Response to Reviewer RC4T [1/7]**
>
> We would like to sincerely thank the reviewer for their thorough and thoughtful feedback.
> Your comments have been instrumental in helping us improve the quality of our work, both by addressing specific concerns and by identifying areas that required further exploration.
> In response, we have conducted additional experiments and analyses to provide more comprehensive evidence and insights.
> Below, we detail our replies to each of your concerns and highlight the changes made.
>
> ---
>
> > In retrieval tasks, there is a significant performance gap compared to black-box models, and the performance is also lower than BM25. Therefore, additional performance comparisons are needed when applying them to various downstream tasks such as sentiment classification and retrieval.
>
> Thank you for your valuable suggestions. We agree that evaluating additional downstream tasks is crucial for thoroughly demonstrating the performance of our framework.
>
> In addition to the STS, retrieval, and clustering tasks, we have also tested our framework on four additional downstream tasks from the MTEB benchmark: classification, pair classification, reranking, and summarization evaluation. The results are provided in **Tables 1–4** below.
>
> As shown in **Tables 1–4**, our framework consistently outperforms existing interpretable text embedding models and achieves performance comparable to many advanced black-box models across all tested downstream tasks. These results further highlight the generalizability and robustness of our framework for diverse text embedding tasks.

---

> ### Author Response · Authors · 2024-11-25
> **Response to Reviewer RC4T [2/7]**
>
> **Table 1: the Classification Task with 12 Datasets**
>
> | Model     | AmazonCounterfactualClassification | AmazonPolarityClassification | AmazonReviewsClassification | Banking77Classification |
> | --------------- | ---------------------------------- | ---------------------------- | --------------------------- | ----------------------- |
> | BERT      | 74.25  | 71.33    | 33.56   | 63.41     |
> | SimCSE (Unsup.) | 67.09  | 74.48    | 33.85   | 73.55     |
> | GloVe     | 56.91  | 60.32    | 29.67   | 67.69     |
> | SimCSE (Sup.)   | 75.75  | 82.47    | 39.6    | 75.76     |
> | SBERT (New)     | 65.28  | 62.98    | 30.79   | 80.4      |
> | AnglE     | 75.55  | 92.84    | 48.29   | 87.69     |
> | OpenAI    | 75.94  | 86.72    | 44.78   | 80.66     |
> | BoT       | 78.87  | 55.28    | 27.95   | 60.63     |
> | QAEmb-MBQA      | 59.81  | 84.43    | 40.31   | 77.72     |
> | CQG-MBQA  | 62.62  | 93.66    | 45.39   | 83.45     |
>
> | Model     | EmotionClassification | ImdbClassification | MassiveIntentClassification | MassiveScenarioClassification |
> |-----------------|-----------------------|--------------------|-----------------------------|-------------------------------|
> | BERT      | 35.28   | 65.35  | 59.88   | 64.28     |
> | SimCSE (Unsup.) | 42.22   | 69.63  | 59.84   | 66.25     |
> | GloVe     | 36.93   | 62.57  | 56.19   | 66.03     |
> | SimCSE (Sup.)   | 44.81   | 73.53  | 65.95   | 70.78     |
> | SBERT (New)     | 41.17   | 59.76  | 67.15   | 74.58     |
> | AnglE     | 51.75   | 92.78  | 76.5    | 79.75     |
> | OpenAI    | 48.74   | 77.98  | 70.15   | 75.33     |
> | BoT       | 22.17   | 53.32  | 48.79   | 49.63     |
> | QAEmb-MBQA      | 39.68   | 89.27  | 62.52   | 68.87     |
> | CQG-MBQA  | 46.04   | 92.8 | 70.2    | 74.9      |
>
>
> | Model     | MTOPDomainClassification | MTOPIntentClassification | ToxicConversationsClassification | TweetSentimentExtractionClassification | Avg.  |
> | --------------- | ------------------------ | ------------------------ | -------------------------------- | -------------------------------------- | ----- |
> | BERT      | 82.63      | 68.14      | 70   | 51.81      | 61.66 |
> | SimCSE (Unsup.) | 81.71      | 59.23      | 68.82  | 53.36      | 62.50 |
> | GloVe     | 79.11      | 55.85      | 65.4 | 50.8 | 57.29 |
> | SimCSE (Sup.)   | 84.29      | 63.14      | 72.04  | 59.73      | 67.32 |
> | SBERT (New)     | 91.9 | 62.84      | 67.47  | 54.25      | 63.21 |
> | AnglE     | 94.02      | 76.92      | 71.09  | 59.75      | 75.58 |
> | OpenAI    | 92.13      | 64.68      | 72.29  | 61.81      | 70.93 |
> | BoT       | 72.77      | 58.41      | 53.24  | 43.59      | 52.05 |
> | QAEmb-MBQA      | 80.95      | 60.23      | 59.91  | 56.03      | 64.98 |
> | CQG-MBQA  | 89.79      | 66.95      | 60.79  | 61.48      | 70.67 |
>
>
>
> **Table 2: the Pair Classification Task with 3 Datasets**
> | Model     | SprintDuplicateQuestions | TwitterSemEval2015 | TwitterURLCorpus | Avg.  |
> | --------------- | ------------------------ | ------------------ | ---------------- | ----- |
> | BERT      | 36.81      | 55.9 | 76.29      | 56.33 |
> | SimCSE (Unsup.) | 78.03      | 61.01  | 81.37      | 73.47 |
> | GloVe     | 86.96      | 53.12  | 77.35      | 72.48 |
> | SimCSE (Sup.)   | 73.04      | 67.75  | 83.89      | 74.89 |
> | SBERT (New)     | 92.58      | 70.02  | 84.77      | 82.46 |
> | AnglE     | 97.24      | 78.17  | 86.33      | 87.25 |
> | OpenAI    | 92.17      | 75.28  | 87.22      | 84.89 |
> | BoT       | 83.33      | 59.82  | 78.63      | 73.26 |
> | QAEmb-MBQA      | 43.71      | 60.04  | 73.21      | 59.65 |
> | CQG-MBQA  | 81.77      | 67.42  | 79.13      | 76.11 |
>
>
>
> **Table 3: the Reranking Task with 3 Datasets**
> | Model     | AskUbuntuDupQuestions | MindSmallReranking | SciDocsRR | StackOverflowDupQuestions | Avg.  |
> | --------------- | --------------------- | ------------------ | --------- | ------------------------- | ----- |
> | BERT      | 45.84   | 28.37  | 64.94     | 34.62 | 43.44 |
> | SimCSE (Unsup.) | 51.57   | 28.62  | 66.33     | 39.35 | 46.47 |
> | GloVe     | 49.57   | 27.01  | 62.56     | 34.03 | 43.29 |
> | SimCSE (Sup.)   | 51.8    | 29.3 | 70.14     | 38.9  | 47.54 |
> | SBERT (New)     | 64.06   | 31.02  | 87.2      | 51.47 | 58.44 |
> | AnglE     | 64.2    | 32.51  | 87.49     | 55.32 | 59.88 |
> | OpenAI    | 62.05   | 31.45  | 81.22     | 50.54 | 56.32 |
> | BoT       | 49.28   | 23.99  | 56.20     | 37.99 | 41.86 |
> | QAEmb-MBQA      | 54.70   | 28.73  | 70.86     | 40.81 | 48.78 |
> | CQG-MBQA  | 59.61   | 30.83  | 81.72     | 47.33 | 54.87 |
>
> **Table 4: the Summarization Task with 1 Dataset**
> | Model     | SummEval |
> | --------------- | -------- |
> | BERT      | 29.82    |
> | SimCSE (Unsup.) | 31.15    |
> | GloVe     | 28.87    |
> | SimCSE (Sup.)   | 31.17    |
> | SBERT (New)     | 27.9     |
> | AnglE     | 32.03    |
> | OpenAI    | 30.8     |
> | BoT       | 28.2     |
> | QAEmb-MBQA      | 28.57    |
> | CQG-MBQA  | 30.41    |

---

> ### Author Response · Authors · 2024-11-25
> **Response to Reviewer RC4T [3/7]**
>
> ---
>
> > Lack of ablation studies to assess the efficacy of the proposed approach
> > * lack of comparison between different models in Figure 4 and 5, and lack of comparison between the MBQA method and directly using the LLM’s outputs.
> > * comparison between vanilla CQG with positive and hard/easy negative, and CQG with positive and negative samples
> > * comparison between having and not having the probing mechanism to refine the generated questions
>
> We conducted extensive ablation studies to address the specific sub-points raised:
>
> 1. **Comparison between QAEmb-MBQA and CQG-MBQA under varying dimensions and thresholds**
>
> In **Tables 5-8**, we present a tabular version of the results shown in Figures 4 and 5.
> Our results reveal that the performance of both CQG-MBQA and QAEmb-MBQA improves as the number of dimensions increases from **1000** to **3000**, stabilizing beyond **3000** dimensions.
> Notably, CQG-MBQA consistently outperforms QAEmb-MBQA across all dimensionalities and binary classification thresholds while maintaining a lower cognitive load.
>
> In the revised version of the paper (to be submitted by the end of November 27), we will incorporate QAEmb-MBQA into **Figures 4 and 5** using data from **Tables 5-8**.
> These additional results will provide a more comprehensive comparison with QAEmb-MBQA and further enhance the clarity and depth of our analysis.
>
> **Table 5: Spearman Correlation vs. the Number of Dimensions**
>
> | Model | # Dimensions |STS12 | STS13 | STS14 | STS15 | STS16 | STSBenchmark | SICK-R |
> |---|---|---|---|---|---|---|---|---|
> | CQG-MBQA | 1000 |63.02 | 77.08 | 69.77 | 77.92 | 75.25 | 78.92 | 74.30 |
> | CQG-MBQA | 2000 |67.08 | 79.74 | 72.71 | 79.28 | 76.84 | 80.74 | 76.08 |
> | CQG-MBQA | 3000 |67.65 | 79.97 | 73.44 | 80.10 | 77.41 | 81.35 | 76.91 |
> | CQG-MBQA | 4000 |68.84 | 80.29 | 73.69 | 79.80 | 78.02 | 81.94 | 77.34 |
> | CQG-MBQA | 5000 |69.28 | 80.00 | 73.60 | 79.87 | 77.82 | 82.20 | 77.51 |
> | CQG-MBQA | 6000 |69.15 | 79.95 | 73.74 | 79.84 | 77.93 | 82.35 | 77.82 |
> | CQG-MBQA | 7000 |69.21 | 80.12 | 73.76 | 80.01 | 77.97 | 82.52 | 78.09 |
> | CQG-MBQA | 8000 |69.06 | 79.95 | 73.75 | 80.36 | 78.14 | 82.51 | 78.20 |
> | CQG-MBQA | 9000 |69.13 | 80.03 | 73.91 | 80.56 | 78.25 | 82.60 | 78.28 |
> | QAEmb-MBQA | 1000 |59.80 | 63.63 | 57.75 | 68.67 | 63.08 | 70.80 | 71.81 |
> | QAEmb-MBQA | 2000 |59.72 | 62.44 | 56.77 | 67.84 | 61.89 | 70.02 | 71.79 |
> | QAEmb-MBQA | 3000 |59.23 | 62.66 | 57.16 | 68.52 | 62.55 | 70.26 | 72.07 |
> | QAEmb-MBQA | 4000 |59.49 | 62.67 | 57.17 | 68.38 | 62.63 | 70.42 | 72.05 |
> | QAEmb-MBQA | 5000 |59.48 | 63.09 | 57.55 | 68.71 | 62.77 | 70.76 | 72.09 |
> | QAEmb-MBQA | 6000 |59.52 | 63.18 | 57.71 | 68.73 | 62.83 | 71.05 | 72.30 |
> | QAEmb-MBQA | 7000 |59.55 | 63.29 | 57.79 | 68.94 | 63.04 | 71.23 | 72.39 |
> | QAEmb-MBQA | 8000 |59.52 | 63.35 | 57.82 | 69.17 | 63.21 | 71.29 | 72.31 |
> | QAEmb-MBQA | 9000 |59.39 | 63.21 | 57.69 | 69.14 | 63.11 | 71.20 | 72.28 |
>
>
> **Table 6: Cognitive Load vs. the Number of Dimensions**
>
> | Model | # Dimensions |STS12 | STS13 | STS14 | STS15 | STS16 | STSBenchmark | SICK-R |
> |---|---|---|---|---|---|---|---|---|
> | CQG-MBQA | 1000 |49 | 46 | 48 | 45 | 50 | 47 | 43 |
> | CQG-MBQA | 2000 |101 | 93 | 98 | 91 | 102 | 95 | 89 |
> | CQG-MBQA | 3000 |152 | 140 | 146 | 137 | 154 | 143 | 132 |
> | CQG-MBQA | 4000 |203 | 186 | 195 | 183 | 204 | 190 | 178 |
> | CQG-MBQA | 5000 |258 | 236 | 246 | 232 | 259 | 240 | 225 |
> | CQG-MBQA | 6000 |305 | 279 | 291 | 273 | 305 | 284 | 265 |
> | CQG-MBQA | 7000 |354 | 323 | 337 | 315 | 353 | 329 | 307 |
> | CQG-MBQA | 8000 |408 | 373 | 388 | 361 | 406 | 378 | 349 |
> | CQG-MBQA | 9000 |453 | 413 | 431 | 400 | 449 | 420 | 388 |
> | QAEmb-MBQA | 1000 |156 | 152 | 155 | 135 | 151 | 133 | 95 |
> | QAEmb-MBQA | 2000 |312 | 303 | 312 | 273 | 301 | 266 | 186 |
> | QAEmb-MBQA | 3000 |462 | 449 | 462 | 406 | 446 | 400 | 287 |
> | QAEmb-MBQA | 4000 |603 | 585 | 604 | 531 | 584 | 520 | 372 |
> | QAEmb-MBQA | 5000 |757 | 732 | 755 | 666 | 733 | 653 | 471 |
> | QAEmb-MBQA | 6000 |904 | 873 | 901 | 797 | 879 | 778 | 556 |
> | QAEmb-MBQA | 7000 |1060 | 1022 | 1057 | 935 | 1029 | 914 | 652 |
> | QAEmb-MBQA | 8000 |1220 | 1176 | 1216 | 1079 | 1183 | 1052 | 756 |
> | QAEmb-MBQA | 9000 |1378 | 1331 | 1376 | 1220 | 1335 | 1190 | 855 |

---

> ### Author Response · Authors · 2024-11-25
> **Response to Reviewer RC4T [4/7]**
>
> **Table 7: Spearman Correlation vs. the Binary Classification Threshold**
>
> | Model | Threshold |STS12 | STS13 | STS14 | STS15 | STS16 | STSBenchmark | SICK-R |
> |---|---|---|---|---|---|---|---|---|
> | CQG-MBQA | 0.1 |72.66 | 81.66 | 75.98 | 81.91 | 80.95 | 83.96 | 80.45 |
> | CQG-MBQA | 0.2 |71.67 | 81.00 | 75.18 | 81.11 | 79.74 | 83.29 | 79.85 |
> | CQG-MBQA | 0.3 |70.64 | 80.68 | 74.48 | 80.81 | 79.01 | 82.88 | 79.28 |
> | CQG-MBQA | 0.4 |69.85 | 80.31 | 74.29 | 80.99 | 78.56 | 82.79 | 78.66 |
> | CQG-MBQA | 0.5 |69.21 | 80.19 | 73.91 | 80.66 | 78.30 | 82.69 | 78.21 |
> | CQG-MBQA | 0.6 |68.28 | 79.65 | 73.01 | 80.43 | 78.01 | 82.26 | 77.42 |
> | CQG-MBQA | 0.7 |67.05 | 78.79 | 72.04 | 80.55 | 77.17 | 81.86 | 76.76 |
> | CQG-MBQA | 0.8 |65.43 | 77.80 | 70.62 | 80.27 | 76.76 | 80.92 | 75.85 |
> | CQG-MBQA | 0.9 |63.46 | 73.88 | 66.98 | 80.00 | 74.51 | 79.54 | 74.56 |
> | QAEmb-MBQA | 0.1 |64.54 | 66.86 | 62.11 | 70.65 | 72.17 | 75.04 | 75.31 |
> | QAEmb-MBQA | 0.2 |63.82 | 65.60 | 60.11 | 69.80 | 68.82 | 73.08 | 74.52 |
> | QAEmb-MBQA | 0.3 |62.16 | 64.51 | 59.18 | 69.37 | 65.76 | 72.19 | 73.50 |
> | QAEmb-MBQA | 0.4 |60.65 | 63.82 | 58.56 | 69.30 | 64.02 | 71.85 | 72.78 |
> | QAEmb-MBQA | 0.5 |59.40 | 63.19 | 57.68 | 69.29 | 63.18 | 71.33 | 72.33 |
> | QAEmb-MBQA | 0.6 |57.92 | 62.94 | 56.99 | 69.62 | 62.59 | 70.70 | 71.92 |
> | QAEmb-MBQA | 0.7 |56.68 | 62.50 | 55.86 | 69.52 | 62.38 | 69.71 | 71.73 |
> | QAEmb-MBQA | 0.8 |54.94 | 61.44 | 53.86 | 68.98 | 62.16 | 69.01 | 71.42 |
> | QAEmb-MBQA | 0.9 |51.84 | 59.53 | 50.51 | 67.71 | 61.25 | 67.70 | 69.74 |
>
> **Table 8: Cognitive Load vs. the Binary Classification Threshold**
>
> | Model | Threshold |STS12 | STS13 | STS14 | STS15 | STS16 | STSBenchmark | SICK-R |
> |---|---|---|---|---|---|---|---|---|
> | CQG-MBQA | 0.1 |1814 | 1652 | 1749 | 1683 | 1776 | 1784 | 1790 |
> | CQG-MBQA | 0.2 |1139 | 1034 | 1092 | 1035 | 1120 | 1088 | 1053 |
> | CQG-MBQA | 0.3 |832 | 756 | 796 | 749 | 821 | 783 | 742 |
> | CQG-MBQA | 0.4 |633 | 576 | 604 | 566 | 626 | 591 | 552 |
> | CQG-MBQA | 0.5 |481 | 439 | 458 | 426 | 478 | 446 | 413 |
> | CQG-MBQA | 0.6 |373 | 341 | 354 | 328 | 372 | 345 | 318 |
> | CQG-MBQA | 0.7 |280 | 257 | 265 | 246 | 281 | 260 | 240 |
> | CQG-MBQA | 0.8 |196 | 181 | 185 | 173 | 197 | 182 | 170 |
> | CQG-MBQA | 0.9 |107 | 101 | 102 | 96 | 109 | 100 | 97 |
> | QAEmb-MBQA | 0.1 |4789 | 4450 | 4729 | 4342 | 4559 | 4620 | 4404 |
> | QAEmb-MBQA | 0.2 |3275 | 3047 | 3234 | 2882 | 3111 | 2999 | 2611 |
> | QAEmb-MBQA | 0.3 |2526 | 2371 | 2499 | 2212 | 2412 | 2249 | 1828 |
> | QAEmb-MBQA | 0.4 |2025 | 1924 | 2011 | 1780 | 1947 | 1773 | 1359 |
> | QAEmb-MBQA | 0.5 |1626 | 1571 | 1625 | 1443 | 1577 | 1408 | 1018 |
> | QAEmb-MBQA | 0.6 |1302 | 1289 | 1313 | 1176 | 1280 | 1124 | 772 |
> | QAEmb-MBQA | 0.7 |1015 | 1039 | 1038 | 942 | 1017 | 882 | 576 |
> | QAEmb-MBQA | 0.8 |730 | 787 | 762 | 705 | 754 | 645 | 400 |
> | QAEmb-MBQA | 0.9 |409 | 482 | 443 | 422 | 451 | 376 | 219 |
>
> 2. **Comparison between MBQA and directly using the LLM’s outputs**
>
> Thank you for your suggestion.
> Conducting experiments using LLM output on the STS datasets poses significant computational and financial challenges.
> Specifically, it would require approximately **47.7 million** API calls, even if we group 20 questions into a single prompt. This would entail thousands of dollars in API credits (assuming 300 tokens per API call) or at least several months of computation time using a local LLM (assuming each API call takes 0.1 seconds).
>
> As an alternative, we have evaluated the accuracy of our MBQA model in replicating LLM outputs.
> In **Table 6** of the appendix in our original paper, we demonstrate that the classification accuracy is **96%**, indicating that the MBQA model is sufficiently accurate when compared to the LLM output.
> Furthermore, the strong performance of our model on seven downstream tasks (three reported in our original paper and four additional tasks shown in **Tables 1–4**) provides indirect evidence of the MBQA model's competitiveness and reliability.
>
> We hope this addresses your concern and clarifies the robustness of our approach.

---

> ### Author Response · Authors · 2024-11-25
> **Response to Reviewer RC4T [5/7]**
>
> 3. **Comparison between vanilla CQG with different types of samples:**
>
> Thank you for your suggestion. We conducted ablation studies to evaluate the impact of different types of samples on the CQG method's performance.
> Specifically, we tested three scenarios, with results summarized in **Table 9** below:
>
> - **(a) Only positive samples:**
>   - We modified the question generation prompt to exclude explicit negative samples:
>     - **Original Prompt:**
>       `...where for all questions, the answer will be "yes" for ALL the positive articles and "no" for ALL the negative articles.`
>     - **Modified Prompt:**
>       `...where for all questions, the answer will be "yes" for ALL the positive articles and "no" for general articles.`
>   - Results show that explicit negative samples improve performance from **76.57** to **77.60** (average across STS datasets).
>
> - **(b) No hard negatives:**
>   - We set the hard negative samples per cluster ($n_h$) and the hard negative probe samples per question ($p_h$) to **0**.
>   - Results show performance drops to **76.26**, compared to **77.60** with both hard and easy negatives included.
>
> - **(c) No easy negatives:**
>   - We set the easy negative samples per cluster ($n_e$) and the easy negative probe samples per question ($p_e$) to **0**.
>   - Results show performance drops to **75.26**, compared to **77.60** with both hard and easy negatives included.
>
> From **Table 9**, it is evident that the full CQG-MBQA method (including positive samples, hard negatives, and easy negatives) yields the highest question quality and achieves the best downstream performance on the STS datasets.
>
> **Table 9: Spearman Correlation on STS Datasets for Different Types of Samples**
> | Model       | STS12 | STS13 | STS14 | STS15 | STS16 | STS-B | SICK-R | Avg.  |
> | ----------------- | ----- | ----- | ----- | ----- | ----- | ----- | ------ | ----- |
> | Implicit Negative | 67.67 | 78.58 | 72.48 | 79.24 | 78.64 | 82.13 | 77.24  | 76.57 |
> | No Hard Negative  | 66.73 | 77.14 | 70.48 | 78.77 | 76.21 | 81.07 | 76.44  | 75.26 |
> | No Easy Negative  | 68.90 | 76.12 | 73.17 | 79.63 | 75.08 | 81.59 | 79.34  | 76.26 |
> | CQG-MBQA (Full)   | 69.21 | 80.19 | 73.91 | 80.66 | 78.30 | 82.69 | 78.21  | 77.60 |
>
> 4. **Comparison of CQG with and without the probing mechanism**
>
> As suggested, we conducted ablation studies to evaluate the effect of the probing mechanism on performance.
> For this experiment, we removed the probing mechanism and used the original LLM-generated order.
> Results in **Table 10** below indicate a performance drop to **76.01**, compared to **77.60** when the probing mechanism is included.
>
> **Table 10: Spearman Correlation on STS Datasets with/without the Probing Mechanism**
>
> | Model       | STS12 | STS13 | STS14 | STS15 | STS16 | STS-B | SICK-R | Avg.  |
> | ----------------- | ----- | ----- | ----- | ----- | ----- | ----- | ------ | ----- |
> | CQG-MBQA without Probing  | 68.29 | 77.92 | 71.17 | 79.80 | 77.06 | 81.33 | 76.52  | 76.01 |
> | CQG-MBQA with Probing     | 69.21 | 80.19 | 73.91 | 80.66 | 78.30 | 82.69 | 78.21  | 77.60 |
>
> ---
>
> > Also, because the cognitive load is defined using the dot product, this measure would be directly influenced by the total number of questions. A normalized version (e.g., dot product divided by number of questions) would provide a fairer comparison across different interpretable models in Table 5.
>
> Thank you for your valuable suggestion.
> We have provided **Table 11** below, which shows the normalized cognitive load (in percentage) for the STS task. These normalized results indicate that our CQG-MBQA framework consistently achieves a lower cognitive load compared to the QAEmb-MBQA method across different datasets. Furthermore, the observed trend aligns closely with the results presented in Table 5 of our original paper.
>
> In the revised version of the paper (to be submitted by the end of November 27), we will update Table 5 to include this additional information.
>
> **Table 11: The Normalized Cognitive Load (in percentage) for the STS task**
> | Model   |   STS12 |   STS13 |   STS14 |   STS15 |   STS16 |   STS-B |   SICK-R |   Avg. |
> |:--------------|--------:|--------:|--------:|--------:|--------:|--------:|---------:|-------:|
> | Bag-of-Tokens |    0.03 |    0.01 |    0.02 |    0.02 |    0.03 |    0.02 |     0.02 |   0.02 |
> | QAEmb-MBQA    |   15.26 |   14.75 |   15.25 |   13.54 |   14.80 |   13.22 |     9.56 |  13.77 |
> | CQG-MBQA      |    5.00 |    4.57 |    4.76 |    4.43 |    4.97 |    4.64 |     4.30 |   4.67 |

---

> ### Author Response · Authors · 2024-11-25
> **Response to Reviewer RC4T [6/7]**
>
> ---
>
> > Having cost analysis would be beneficial as MBQA requires significant LLM inferences (or API calls) during training time and even more may be required during Post-processing (probing).
>
> Thank you for your insightful suggestion. We have summarized the incurred costs in **Table 12** below.
>
> In comparison to the cost of generating question-answer pairs for training, the probing process incurs negligible expenses. Additionally, when compared to the costs of generating question-answer pairs for inference directly using `GPT-4o-mini` (as shown in Table 1 of our original paper), the cost of generating question-answer pairs for MBQA training is significantly lower.
>
> **Table 12: The Cost Analysis for Different Steps in CQG-MBQA (using `GPT-4o-mini`)**
> | Stage | Actual Cost (In USD) |
> | ------------------- | -------------------- |
> | Question Generation | 2.52   |
> | Probing       | 1.92   |
> | Question Answers Generation for Training | 30.06  |
>
> ---
>
> > Including a case study on bad examples would also be beneficial—for instance, displaying cases where two texts result in similar embeddings even when those two texts do not necessarily have similar semantics. Are they completely off? Or how could one improve your approach?
>
> Thank you for this valuable suggestion. We have analyzed a challenging case from the STS Benchmark dataset, detailed below:
>
> - **Text Pair:**
>   - **Text 1:** `And that is happening in GOP-controlled states.`
>   - **Text 2:** `Michigan IS a GOP-controlled state.`
> - **Ground Truth Similarity Score (0-5):** `0.8`
> - **CQG-MBQA Prediction (0-1):** `0.851`
>
>
> This pair is labeled as having **low similarity** (0.8 on a [0, 5] scale) in the dataset; however, our model outputs a relatively **high similarity** score (0.851 on a [0, 1] scale). The discrepancy can be attributed to the following factors:
>
> 1. **Incorrect Prediction of QA Answers:**
>    For instance, in dimension 94, the question `Is there an emphasis on the Democratic Party's dynamics?` was incorrectly answered as "yes" for both texts. The correct answer should be "no" for both, as the GOP refers to the Grand Old Party (Republican Party), not the Democratic Party.
>
> 2. **Overly General Questions:**
>    In dimension 209, the question `Is the article set in the United States or Canada?` was correctly answered as "yes" for both texts. However, this question is too general to capture the nuanced differences between the two texts. Specifically:
>    - Text 1 discusses an unspecified event occurring in GOP-controlled states.
>    - Text 2 states a factual assertion about Michigan being a GOP-controlled state.
>
>    Such general questions dilute the embedding’s ability to capture subtle distinctions, leading to an inflated similarity score.
>
> We will continue to investigate this issue and explore potential improvements to enhance the robustness of our approach in future work.

---

> ### Author Response · Authors · 2024-11-25
> **Response to Reviewer RC4T [7/7]**
>
> ---
>
> > Did you evaluate your model on the fMRI task presented in the QAEmb paper for a direct performance comparison?
>
> Thank you for your suggestion. We did not evaluate the fMRI task for the following reasons:
>
> 1. The fMRI task is highly specialized within the neuroscience domain and requires a task-specific design with a relatively small number of questions (e.g., 29 questions as reported in the QAEmb paper).
> 2. None of the authors possess the domain expertise necessary to conduct a thorough evaluation for this task.
>
> Moreover, there are fundamental differences in the experimental settings: QAEmb focuses on task-specific text embeddings using task labels, whereas our framework is designed to produce general-purpose text embeddings, similar to pre-trained text encoders.
> Inspired by QAEmb, we recreated the QAEmb baseline (QAEmb-MBQA) under a general text embedding setting for fair comparison.
>
> Instead, we demonstrate the generalizability of our framework through evaluations on **7** diverse downstream tasks from the MTEB benchmark, as presented in our original paper and summarized in **Tables 1–4** in this response.
>
> ---
>
> > How do variations in the number of initial clusters (k) or the use of different clustering methods affect the performance of CQG?
>
> We selected $k=5000$ based on the elbow point observed in the MSE vs. number of clusters plot. In the revised version of the paper (to be submitted by November 27), we will include this figure in the appendix to provide further clarity.
>
> We agree that the number of clusters ($k$) significantly impacts performance, as it determines the granularity of the positive and negative sample boundaries. However, evaluating the framework with different values of $k$ and exploring alternative clustering methods requires rerunning the entire set of experiments multiple times, which would take several weeks.
>
> Due to time constraints, we were unable to conduct comprehensive studies on $k$ or alternative clustering methods. Nevertheless, we agree that exploring this aspect further, such as testing other clustering algorithms (e.g., DBSCAN or hierarchical clustering), is an excellent direction for future research.
>
> ---
>
> We hope our detailed responses and additional analyses address your concerns and provide clarity on our contributions and methodology. Your feedback has been invaluable in improving the robustness and generalizability of our work. Thank you again for your time and effort in reviewing our paper, and we look forward to your reply.

---

> > ### Comment · Reviewer_RC4T · 2024-11-28
> > **Response acknowledgement**
> >
> > Thanks for all of your responses, I can see that authors have put a lot of effort in preparing the response and most of my questions are clarified.
> >
> > I would like to keep my score as is, as my scoring was based on my understanding of the novelty and contribution of this paper, without fine-grained deduction of points related to the weakness or questions I raised.
> > I hope that experimental results on more downstream tasks can help support the generalizability of the proposed framework, and the suggested analysis of the challenging example can be a foundation for further development of this framework.

---

> > > ### Author Response · Authors · 2024-11-28
> > >
> > > Thank you for your thoughtful and constructive feedback throughout the review process. While we understand and respect your decision to maintain the score, we want to assure you that we have carefully polished the paper based on your valuable feedback. Please do not hesitate to reach out if any further questions or concerns arise.

---

### Official Review · Reviewer_sYsh · 2024-11-04

**Soundness:** 4
**Presentation:** 3
**Contribution:** 3
**Rating:** 8
**Confidence:** 4

**Summary:**

This paper introduces CQG-MBQA (Contrastive Question Generation - Multi-task Binary Question Answering), a general framework to create interpretable semantic text embeddings for NLP tasks. This framework emphasizes interpretability, which is essential for tasks requiring transparency, such as legal and medical applications. Traditional black-box text embedding methods, while effective, lack interpretability, limiting their utility in such cases. By comparison, CQG-MBQA is able to generate interpretable semantic text embeddings via binary yes/no questions. To be concrete, this framework first generates binary yes/no questions through contrastive question generation (CQG) using GPT-4o-mini for the entire corpus. Then, it fine-tunes a multi-task binary question-answering (MBQA) model by distilling knowledge from GPT-4o-mini. In this way, one can use MBQA to create the interpretable embeddings for text, without relying on LLMs, thus reducing the API costs. The experimental results show that CQG-MBQA performs comparably to advanced black-box models, and outperforms other interpretable text embedding models across various downstream tasks.

**Strengths:**

1. CQG-MBQA focus on producing interpretable embeddings, which is important for domains requiring transparency.
2. Compared with QAEmb,  CQG produces more discriminative questions.
3. By integrating MBQA, the framework achieves cost-effective embeddings compared to LLM-based alternatives.
4. This paper conducts extensive experiments on semantic textual similarity, retrieval, and clustering tasks, showcasing its utility and competitiveness.

**Weaknesses:**

Please refer to the Questions.

**Questions:**

1. Does the CQG-MBQA framework need to generate a new set of yes/no questions every time it is applied to a different dataset? If so, is there a way to enhance the generalizability of the CQG-MBQA model? In other words, could a more universal set of yes/no questions be designed to handle multiple tasks/datasets, rather than creating a separate set tailored to each specific task/dataset?

2. Figure 4 shows that with around 3,000 questions, CQG-MBQA can achieve high-quality text embeddings on STS tasks, and using additional questions does not improve embedding quality; instead, it decreases interpretability. Does this imply that the yes/no questions generated by CQG have semantic overlap or inclusion relationships? In other words, is there a large number of semantically similar questions within the set of yes/no questions?

---

> ### Author Response · Authors · 2024-11-25
> **Response to Reviewer sYsh [1/3]**
>
> We sincerely thank the reviewer for these insightful and thought-provoking questions. They address key aspects of our framework's generalizability, design choices, and practical implications.
> Below, we provide detailed responses to each question, supported by additional experiments and clarifications to address the raised concerns comprehensively.
>
> ---
> > Question 1: Does the CQG-MBQA framework need to generate a new set of yes/no questions every time it is applied to a different dataset? If so, is there a way to enhance the generalizability of the CQG-MBQA model? In other words, could a more universal set of yes/no questions be designed to handle multiple tasks/datasets, rather than creating a separate set tailored to each specific task/dataset?
>
> Thank you for raising this insightful question. We want to clarify that the CQG-MBQA framework does **not** require generating a new set of questions for each dataset. In our experiments, we generated a universal set of questions using the CQG method on the MEDI2 dataset, trained the MBQA model on these questions, and tested the same model across multiple datasets from various downstream tasks without additional training or fine-tuning. The superior results across diverse downstream tasks demonstrate the model's generalizability.
>
> However, our framework can allow users to generate task-specific questions using their own text corpus. This optional customization requires no labeled training data and can better suit specific needs. For example, questions generated from hotel customer reviews could help uncover customer preferences across different hotel types.
>
> Our experiments show that the pre-trained model, using questions derived from the MEDI2 dataset, performs robustly on three downstream tasks (STS, retrieval, and clustering) as reported in our original paper, without requiring further question generation.
> To further validate this generalizability, we conducted experiments on four additional downstream tasks from the MTEB benchmark.
> **Tables 1–4** below highlight how our framework consistently outperforms existing interpretable text embedding models and achieves comparable performance to many advanced black-box models across all tested downstream tasks.
>
> In the revised version of the paper (to be submitted by the end of November 27), we will clarify this point in the main paper and include the results from **Tables 1–4** in the supplementary materials.

---

> ### Author Response · Authors · 2024-11-25
> **Response to Reviewer sYsh [2/3]**
>
> **Table 1: the Classification Task with 12 Datasets**
>
> | Model     | AmazonCounterfactualClassification | AmazonPolarityClassification | AmazonReviewsClassification | Banking77Classification |
> | --------------- | ---------------------------------- | ---------------------------- | --------------------------- | ----------------------- |
> | BERT      | 74.25  | 71.33    | 33.56   | 63.41     |
> | SimCSE (Unsup.) | 67.09  | 74.48    | 33.85   | 73.55     |
> | GloVe     | 56.91  | 60.32    | 29.67   | 67.69     |
> | SimCSE (Sup.)   | 75.75  | 82.47    | 39.6    | 75.76     |
> | SBERT (New)     | 65.28  | 62.98    | 30.79   | 80.4      |
> | AnglE     | 75.55  | 92.84    | 48.29   | 87.69     |
> | OpenAI    | 75.94  | 86.72    | 44.78   | 80.66     |
> | BoT       | 78.87  | 55.28    | 27.95   | 60.63     |
> | QAEmb-MBQA      | 59.81  | 84.43    | 40.31   | 77.72     |
> | CQG-MBQA  | 62.62  | 93.66    | 45.39   | 83.45     |
>
> | Model     | EmotionClassification | ImdbClassification | MassiveIntentClassification | MassiveScenarioClassification |
> |-----------------|-----------------------|--------------------|-----------------------------|-------------------------------|
> | BERT      | 35.28   | 65.35  | 59.88   | 64.28     |
> | SimCSE (Unsup.) | 42.22   | 69.63  | 59.84   | 66.25     |
> | GloVe     | 36.93   | 62.57  | 56.19   | 66.03     |
> | SimCSE (Sup.)   | 44.81   | 73.53  | 65.95   | 70.78     |
> | SBERT (New)     | 41.17   | 59.76  | 67.15   | 74.58     |
> | AnglE     | 51.75   | 92.78  | 76.5    | 79.75     |
> | OpenAI    | 48.74   | 77.98  | 70.15   | 75.33     |
> | BoT       | 22.17   | 53.32  | 48.79   | 49.63     |
> | QAEmb-MBQA      | 39.68   | 89.27  | 62.52   | 68.87     |
> | CQG-MBQA  | 46.04   | 92.8 | 70.2    | 74.9      |
>
>
> | Model     | MTOPDomainClassification | MTOPIntentClassification | ToxicConversationsClassification | TweetSentimentExtractionClassification | Avg.  |
> | --------------- | ------------------------ | ------------------------ | -------------------------------- | -------------------------------------- | ----- |
> | BERT      | 82.63      | 68.14      | 70   | 51.81      | 61.66 |
> | SimCSE (Unsup.) | 81.71      | 59.23      | 68.82  | 53.36      | 62.50 |
> | GloVe     | 79.11      | 55.85      | 65.4 | 50.8 | 57.29 |
> | SimCSE (Sup.)   | 84.29      | 63.14      | 72.04  | 59.73      | 67.32 |
> | SBERT (New)     | 91.9 | 62.84      | 67.47  | 54.25      | 63.21 |
> | AnglE     | 94.02      | 76.92      | 71.09  | 59.75      | 75.58 |
> | OpenAI    | 92.13      | 64.68      | 72.29  | 61.81      | 70.93 |
> | BoT       | 72.77      | 58.41      | 53.24  | 43.59      | 52.05 |
> | QAEmb-MBQA      | 80.95      | 60.23      | 59.91  | 56.03      | 64.98 |
> | CQG-MBQA  | 89.79      | 66.95      | 60.79  | 61.48      | 70.67 |
>
> **Table 2: the Pair Classification Task with 3 Datasets**
> | Model     | SprintDuplicateQuestions | TwitterSemEval2015 | TwitterURLCorpus | Avg.  |
> | --------------- | ------------------------ | ------------------ | ---------------- | ----- |
> | BERT      | 36.81      | 55.9 | 76.29      | 56.33 |
> | SimCSE (Unsup.) | 78.03      | 61.01  | 81.37      | 73.47 |
> | GloVe     | 86.96      | 53.12  | 77.35      | 72.48 |
> | SimCSE (Sup.)   | 73.04      | 67.75  | 83.89      | 74.89 |
> | SBERT (New)     | 92.58      | 70.02  | 84.77      | 82.46 |
> | AnglE     | 97.24      | 78.17  | 86.33      | 87.25 |
> | OpenAI    | 92.17      | 75.28  | 87.22      | 84.89 |
> | BoT       | 83.33      | 59.82  | 78.63      | 73.26 |
> | QAEmb-MBQA      | 43.71      | 60.04  | 73.21      | 59.65 |
> | CQG-MBQA  | 81.77      | 67.42  | 79.13      | 76.11 |
>
> **Table 3: the Reranking Task with 3 Datasets**
> | Model     | AskUbuntuDupQuestions | MindSmallReranking | SciDocsRR | StackOverflowDupQuestions | Avg.  |
> | --------------- | --------------------- | ------------------ | --------- | ------------------------- | ----- |
> | BERT      | 45.84   | 28.37  | 64.94     | 34.62 | 43.44 |
> | SimCSE (Unsup.) | 51.57   | 28.62  | 66.33     | 39.35 | 46.47 |
> | GloVe     | 49.57   | 27.01  | 62.56     | 34.03 | 43.29 |
> | SimCSE (Sup.)   | 51.8    | 29.3 | 70.14     | 38.9  | 47.54 |
> | SBERT (New)     | 64.06   | 31.02  | 87.2      | 51.47 | 58.44 |
> | AnglE     | 64.2    | 32.51  | 87.49     | 55.32 | 59.88 |
> | OpenAI    | 62.05   | 31.45  | 81.22     | 50.54 | 56.32 |
> | BoT       | 49.28   | 23.99  | 56.20     | 37.99 | 41.86 |
> | QAEmb-MBQA      | 54.70   | 28.73  | 70.86     | 40.81 | 48.78 |
> | CQG-MBQA  | 59.61   | 30.83  | 81.72     | 47.33 | 54.87 |
>
> **Table 4: the Summarization Task with 1 Dataset**
> | Model     | SummEval |
> | --------------- | -------- |
> | BERT      | 29.82    |
> | SimCSE (Unsup.) | 31.15    |
> | GloVe     | 28.87    |
> | SimCSE (Sup.)   | 31.17    |
> | SBERT (New)     | 27.9     |
> | AnglE     | 32.03    |
> | OpenAI    | 30.8     |
> | BoT       | 28.2     |
> | QAEmb-MBQA      | 28.57    |
> | CQG-MBQA  | 30.41    |

---

> ### Author Response · Authors · 2024-11-25
> **Response to Reviewer sYsh [3/3]**
>
> ---
>
> > Question 2: Figure 4 shows that with around 3,000 questions, CQG-MBQA can achieve high-quality text embeddings on STS tasks, and using additional questions does not improve embedding quality; instead, it decreases interpretability. Does this imply that the yes/no questions generated by CQG have semantic overlap or inclusion relationships? In other words, is there a large number of semantically similar questions within the set of yes/no questions?
>
> Yes, despite our use of the CQG method to generate diverse and discriminative questions, some semantic overlap naturally exists within the generated set.
> To address this, our CQG-MBQA framework incorporates two postprocessing steps, probing and deduplication, to identify the most effective and non-duplicated questions.
> Specifically, during the deduplication step, we ensure that the semantic similarity (measured by cosine similarity) between the question embeddings does not exceed 0.8.
>
> However, even after deduplication, certain semantic relationships remain. For instance, the questions "Is the enhancement of community well-being through environmental policies discussed?" and "Is there a connection made between health and environmental factors?" are not duplicates but are semantically related with differing focal points.
>
> To provide additional insights, **Tables 5 and 6** below present the performance and cognitive load results on STS tasks for both CQG-MBQA and QAEmb-MBQA models across different dimensional settings.
>
> Our results reveal that the performance of both CQG-MBQA and QAEmb-MBQA improves as the number of dimensions increases from **1000** to **3000**, stabilizing beyond **3000** dimensions.
> Despite we have some semantic overlaping between questions, CQG-MBQA consistently outperforms QAEmb-MBQA across all dimensionalities while maintaining a lower cognitive load.
>
> **Table 5: Spearman Correlation vs. the Number of Dimensions**
> | Model | # Dimensions |STS12 | STS13 | STS14 | STS15 | STS16 | STSBenchmark | SICK-R |
> |---|---|---|---|---|---|---|---|---|
> | CQG-MBQA| 1000 |63.02 | 77.08 | 69.77 | 77.92 | 75.25 | 78.92 | 74.30 |
> | CQG-MBQA | 2000 |67.08 | 79.74 | 72.71 | 79.28 | 76.84 | 80.74 | 76.08 |
> | CQG-MBQA | 3000 |67.65 | 79.97 | 73.44 | 80.10 | 77.41 | 81.35 | 76.91 |
> | CQG-MBQA | 4000 |68.84 | 80.29 | 73.69 | 79.80 | 78.02 | 81.94 | 77.34 |
> | CQG-MBQA | 5000 |69.28 | 80.00 | 73.60 | 79.87 | 77.82 | 82.20 | 77.51 |
> | CQG-MBQA | 6000 |69.15 | 79.95 | 73.74 | 79.84 | 77.93 | 82.35 | 77.82 |
> | CQG-MBQA | 7000 |69.21 | 80.12 | 73.76 | 80.01 | 77.97 | 82.52 | 78.09 |
> | CQG-MBQA | 8000 |69.06 | 79.95 | 73.75 | 80.36 | 78.14 | 82.51 | 78.20 |
> | CQG-MBQA | 9000 |69.13 | 80.03 | 73.91 | 80.56 | 78.25 | 82.60 | 78.28 |
> | QAEmb-MBQA | 1000 |59.80 | 63.63 | 57.75 | 68.67 | 63.08 | 70.80 | 71.81 |
> | QAEmb-MBQA | 2000 |59.72 | 62.44 | 56.77 | 67.84 | 61.89 | 70.02 | 71.79 |
> | QAEmb-MBQA | 3000 |59.23 | 62.66 | 57.16 | 68.52 | 62.55 | 70.26 | 72.07 |
> | QAEmb-MBQA | 4000 |59.49 | 62.67 | 57.17 | 68.38 | 62.63 | 70.42 | 72.05 |
> | QAEmb-MBQA | 5000 |59.48 | 63.09 | 57.55 | 68.71 | 62.77 | 70.76 | 72.09 |
> | QAEmb-MBQA | 6000 |59.52 | 63.18 | 57.71 | 68.73 | 62.83 | 71.05 | 72.30 |
> | QAEmb-MBQA | 7000 |59.55 | 63.29 | 57.79 | 68.94 | 63.04 | 71.23 | 72.39 |
> | QAEmb-MBQA | 8000 |59.52 | 63.35 | 57.82 | 69.17 | 63.21 | 71.29 | 72.31 |
> | QAEmb-MBQA | 9000 |59.39 | 63.21 | 57.69 | 69.14 | 63.11 | 71.20 | 72.28 |
>
> **Table 6: Cognitive Load vs. the Number of Dimensions**
> | Model | # Dimensions |STS12 | STS13 | STS14 | STS15 | STS16 | STSBenchmark | SICK-R |
> |---|---|---|---|---|---|---|---|---|
> | CQG-MBQA | 1000 |49 | 46 | 48 | 45 | 50 | 47 | 43 |
> | CQG-MBQA | 2000 |101 | 93 | 98 | 91 | 102 | 95 | 89 |
> | CQG-MBQA | 3000 |152 | 140 | 146 | 137 | 154 | 143 | 132 |
> | CQG-MBQA | 4000 |203 | 186 | 195 | 183 | 204 | 190 | 178 |
> | CQG-MBQA | 5000 |258 | 236 | 246 | 232 | 259 | 240 | 225 |
> | CQG-MBQA | 6000 |305 | 279 | 291 | 273 | 305 | 284 | 265 |
> | CQG-MBQA | 7000 |354 | 323 | 337 | 315 | 353 | 329 | 307 |
> | CQG-MBQA | 8000 |408 | 373 | 388 | 361 | 406 | 378 | 349 |
> | CQG-MBQA | 9000 |453 | 413 | 431 | 400 | 449 | 420 | 388 |
> | QAEmb-MBQA | 1000 |156 | 152 | 155 | 135 | 151 | 133 | 95 |
> | QAEmb-MBQA | 2000 |312 | 303 | 312 | 273 | 301 | 266 | 186 |
> | QAEmb-MBQA | 3000 |462 | 449 | 462 | 406 | 446 | 400 | 287 |
> | QAEmb-MBQA | 4000 |603 | 585 | 604 | 531 | 584 | 520 | 372 |
> | QAEmb-MBQA | 5000 |757 | 732 | 755 | 666 | 733 | 653 | 471 |
> | QAEmb-MBQA | 6000 |904 | 873 | 901 | 797 | 879 | 778 | 556 |
> | QAEmb-MBQA | 7000 |1060 | 1022 | 1057 | 935 | 1029 | 914 | 652 |
> | QAEmb-MBQA | 8000 |1220 | 1176 | 1216 | 1079 | 1183 | 1052 | 756 |
> | QAEmb-MBQA | 9000 |1378 | 1331 | 1376 | 1220 | 1335 | 1190 | 855 |
>
> ---
>
> We hope these responses address your concerns and provide clarity on the flexibility, generalizability, and design choices of our framework. Thank you once again for your invaluable feedback, and we look forward to your thoughts on our clarifications and updates.

---

> > ### Comment · Reviewer_sYsh · 2024-11-27
> > **Response to Authors**
> >
> > Thank you for your response to my questions. My concerns have been resolved, and I have updated the corresponding score.

---

> > > ### Author Response · Authors · 2024-11-28
> > >
> > > Thank you for taking the time to review our responses and for updating the score. We greatly appreciate your thoughtful feedback, which has helped us refine and improve our work. Please do not hesitate to reach out if any further questions or concerns arise.

---

### Official Review · Reviewer_SSgS · 2024-11-05

**Soundness:** 3
**Presentation:** 3
**Contribution:** 3
**Rating:** 8
**Confidence:** 3

**Summary:**

The paper proposes CQG-MVQA, an intepretable embedding framework. The framework generates questions and binary answers about texts, and trains binary classifiers on each question. Each of the prediction to these questions forms a dimension of an interpretable embedding. It is shown that these interpretable achieves decent performance on MTEB and a reduced cognitive load for interpretability compared to baseline methods in interpretable embeddings.

**Strengths:**

1) The question generation component of the framework concerns generating questions that are both discriminative and general. It groups similar texts by clustering for the generation of questions, such that nuanced questions can be asked for each group, as opposed to simple questions in the baseline method. The concept is analogous to leveraging hard negatives in regular training of embedding models.
2) The authors show good understanding at related work; the implementation and the evaluation are sound from the perspective of sentence embeddings.

**Weaknesses:**

1) Performance ablations about setups in the framework can be very interesting although currently missing (e.g., performance across different dimensionality, question difficulties, different encoding models, etc..).
2) Implementation details can be moved more to the main paper as they are mostly in appendices.

**Questions:**

1) In order to achieve performance of regular dense representation models, which aspects of the framework and the implementation details do the authors think worth scaling up? Is there any interesting evidence? e.g., better datasets for generating questions; more questions to form the embedding dimensions, etc..
2) From my understanding in the appendix, the paper uses UAE-large-v1 as the encoding model, why? What happens if the encoding model is some better models - does it help with the performance of the final interpretable embeddings?

---

> ### Author Response · Authors · 2024-11-25
> **Response to Reviewer SSgS [1/3]**
>
> We appreciate the reviewer’s thoughtful feedback and valuable suggestions, which have significantly helped us improve the quality and robustness of our work. In response, we conducted extensive additional experiments and revisions to address the points raised, focusing on performance ablations, implementation details, and encoder model selection. Below, we provide detailed responses to each comment, outlining the changes and insights gained.
>
> ---
>
> > Performance ablations about setups in the framework can be very interesting although currently missing (e.g., performance across different dimensionality, question difficulties, different encoding models, etc..).
>
> Thank you for highlighting this point. We have conducted several ablation studies based on your suggestions:
>
> 1. **Dimensionality**
>
> Figure 4 in our paper (Spearman correlation and cognitive load vs. number of dimensions) demonstrates the performance of our model across varying dimensionalities. To provide additional insights, **Tables 1 and 2** below present the performance and cognitive load results on STS tasks for both CQG-MBQA and QAEmb-MBQA models across different dimensional settings.
>
> Our results reveal that the performance of both CQG-MBQA and QAEmb-MBQA improves as the number of dimensions increases from **1000** to **3000**, stabilizing beyond **3000** dimensions. Notably, CQG-MBQA consistently outperforms QAEmb-MBQA across all dimensionalities while maintaining a lower cognitive load.
>
> **Table 1: Spearman Correlation vs. the Number of Dimensions**
> | Model | # Dimensions |STS12 | STS13 | STS14 | STS15 | STS16 | STSBenchmark | SICK-R |
> |---|---|---|---|---|---|---|---|---|
> | CQG-MBQA| 1000 |63.02 | 77.08 | 69.77 | 77.92 | 75.25 | 78.92 | 74.30 |
> | CQG-MBQA | 2000 |67.08 | 79.74 | 72.71 | 79.28 | 76.84 | 80.74 | 76.08 |
> | CQG-MBQA | 3000 |67.65 | 79.97 | 73.44 | 80.10 | 77.41 | 81.35 | 76.91 |
> | CQG-MBQA | 4000 |68.84 | 80.29 | 73.69 | 79.80 | 78.02 | 81.94 | 77.34 |
> | CQG-MBQA | 5000 |69.28 | 80.00 | 73.60 | 79.87 | 77.82 | 82.20 | 77.51 |
> | CQG-MBQA | 6000 |69.15 | 79.95 | 73.74 | 79.84 | 77.93 | 82.35 | 77.82 |
> | CQG-MBQA | 7000 |69.21 | 80.12 | 73.76 | 80.01 | 77.97 | 82.52 | 78.09 |
> | CQG-MBQA | 8000 |69.06 | 79.95 | 73.75 | 80.36 | 78.14 | 82.51 | 78.20 |
> | CQG-MBQA | 9000 |69.13 | 80.03 | 73.91 | 80.56 | 78.25 | 82.60 | 78.28 |
> | QAEmb-MBQA | 1000 |59.80 | 63.63 | 57.75 | 68.67 | 63.08 | 70.80 | 71.81 |
> | QAEmb-MBQA | 2000 |59.72 | 62.44 | 56.77 | 67.84 | 61.89 | 70.02 | 71.79 |
> | QAEmb-MBQA | 3000 |59.23 | 62.66 | 57.16 | 68.52 | 62.55 | 70.26 | 72.07 |
> | QAEmb-MBQA | 4000 |59.49 | 62.67 | 57.17 | 68.38 | 62.63 | 70.42 | 72.05 |
> | QAEmb-MBQA | 5000 |59.48 | 63.09 | 57.55 | 68.71 | 62.77 | 70.76 | 72.09 |
> | QAEmb-MBQA | 6000 |59.52 | 63.18 | 57.71 | 68.73 | 62.83 | 71.05 | 72.30 |
> | QAEmb-MBQA | 7000 |59.55 | 63.29 | 57.79 | 68.94 | 63.04 | 71.23 | 72.39 |
> | QAEmb-MBQA | 8000 |59.52 | 63.35 | 57.82 | 69.17 | 63.21 | 71.29 | 72.31 |
> | QAEmb-MBQA | 9000 |59.39 | 63.21 | 57.69 | 69.14 | 63.11 | 71.20 | 72.28 |
>
> **Table 2: Cognitive Load vs. the Number of Dimensions**
> | Model | # Dimensions |STS12 | STS13 | STS14 | STS15 | STS16 | STSBenchmark | SICK-R |
> |---|---|---|---|---|---|---|---|---|
> | CQG-MBQA | 1000 |49 | 46 | 48 | 45 | 50 | 47 | 43 |
> | CQG-MBQA | 2000 |101 | 93 | 98 | 91 | 102 | 95 | 89 |
> | CQG-MBQA | 3000 |152 | 140 | 146 | 137 | 154 | 143 | 132 |
> | CQG-MBQA | 4000 |203 | 186 | 195 | 183 | 204 | 190 | 178 |
> | CQG-MBQA | 5000 |258 | 236 | 246 | 232 | 259 | 240 | 225 |
> | CQG-MBQA | 6000 |305 | 279 | 291 | 273 | 305 | 284 | 265 |
> | CQG-MBQA | 7000 |354 | 323 | 337 | 315 | 353 | 329 | 307 |
> | CQG-MBQA | 8000 |408 | 373 | 388 | 361 | 406 | 378 | 349 |
> | CQG-MBQA | 9000 |453 | 413 | 431 | 400 | 449 | 420 | 388 |
> | QAEmb-MBQA | 1000 |156 | 152 | 155 | 135 | 151 | 133 | 95 |
> | QAEmb-MBQA | 2000 |312 | 303 | 312 | 273 | 301 | 266 | 186 |
> | QAEmb-MBQA | 3000 |462 | 449 | 462 | 406 | 446 | 400 | 287 |
> | QAEmb-MBQA | 4000 |603 | 585 | 604 | 531 | 584 | 520 | 372 |
> | QAEmb-MBQA | 5000 |757 | 732 | 755 | 666 | 733 | 653 | 471 |
> | QAEmb-MBQA | 6000 |904 | 873 | 901 | 797 | 879 | 778 | 556 |
> | QAEmb-MBQA | 7000 |1060 | 1022 | 1057 | 935 | 1029 | 914 | 652 |
> | QAEmb-MBQA | 8000 |1220 | 1176 | 1216 | 1079 | 1183 | 1052 | 756 |
> | QAEmb-MBQA | 9000 |1378 | 1331 | 1376 | 1220 | 1335 | 1190 | 855 |

---

> ### Author Response · Authors · 2024-11-25
> **Response to Reviewer SSgS [2/3]**
>
> 2. **Question Difficulties**
>
> The parameters within the Contrastive Question Generation (CQG) method influence the difficulty level of the generated questions by providing varying amounts of information to the LLM during question generation.
>
> As suggested, we have conducted ablation studies to examine how modifications to the CQG method affect performance, enabling us to better understand the impact of question difficulty on the overall results. Specifically, we tested four cases:
>
> - **(a) Only positive samples:**
>   - We modified the question generation prompt to exclude explicit negative samples:
>     - **Original Prompt:**
>       `...where for all questions, the answer will be "yes" for ALL the positive articles and "no" for ALL the negative articles.`
>     - **Modified Prompt:**
>       `...where for all questions, the answer will be "yes" for ALL the positive articles and "no" for general articles.`
>   - Results show that explicit negative samples improve performance from **76.57** to **77.60** (average across STS datasets).
>
> - **(b) No hard negatives:**
>   - We set the hard negative samples per cluster ($n_h$) and the hard negative probe samples per question ($p_h$) to **0**.
>   - Results show performance drops to **76.26**, compared to **77.60** with both hard and easy negatives included.
>
> - **(c) No easy negatives:**
>   - We set the easy negative samples per cluster ($n_e$) and the easy negative probe samples per question ($p_e$) to **0**.
>   - Results show performance drops to **75.26**, compared to **77.60** with both hard and easy negatives included.
>
> - **(d) Without the probing mechanism:**
>   - We removed the probing mechanism and used the original LLM-generated order.
>   - Results show performance drops to **76.01**, compared to **77.60** with the probing mechanism.
>
> From these ablation studies, we demonstrate that the CQG-MBQA (full) method (including positive samples, hard negatives, easy negatives, and the probing mechanism) yields the highest question quality and achieves the best downstream performance on the STS datasets. Please refer to **Table 3** below for the complete results.
>
> **Table 3: Spearman Correlation on STS Datasets for Different Ablated Models**
> | Model             | STS12 | STS13 | STS14 | STS15 | STS16 | STS-B | SICK-R | Avg.  |
> | ----------------- | ----- | ----- | ----- | ----- | ----- | ----- | ------ | ----- |
> | Implicit Negative | 67.67 | 78.58 | 72.48 | 79.24 | 78.64 | 82.13 | 77.24  | 76.57 |
> | No Hard Negative  | 66.73 | 77.14 | 70.48 | 78.77 | 76.21 | 81.07 | 76.44  | 75.26 |
> | No Easy Negative  | 68.90 | 76.12 | 73.17 | 79.63 | 75.08 | 81.59 | 79.34  | 76.26 |
> | No Probing        | 68.29 | 77.92 | 71.17 | 79.80 | 77.06 | 81.33 | 76.52  | 76.01 |
> | CQG-MBQA (Full)   | 69.21 | 80.19 | 73.91 | 80.66 | 78.30 | 82.69 | 78.21  | 77.60 |
>
> 3.**Different Encoders**
>
> Beyond `UAE-Large-V1`, an advanced encoder that ranks highly on the MTEB benchmark, we further evaluated two alternative encoders: `stella_en_400M_v5` and `gte-large-en-v1.5`. Both of these models also rank highly on the MTEB benchmark and have a comparable parameter size (approximately 400M–500M).
>
> A summary of the results on the STS task is provided in **Table 4** below.
> The results indicate that while these alternative encoders perform reasonably well, `UAE-Large-V1` consistently achieves the best performance.
>
> **Table 4: Spearman Correlation on STS Datasets for Different Encoder Models**
> | Model                        | STS12 | STS13 | STS14 | STS15 | STS16 | STS-B | SICK-R | Avg.  |
> | ---------------------------- | ----- | ----- | ----- | ----- | ----- | ----- | ------ | ----- |
> | CQG-MBQA (stella_en_400M_v5) | 54.45 | 75.66 | 64.92 | 76.13 | 74.20 | 74.01 | 73.37  | 70.39 |
> | CQG-MBQA (gte-large-en-v1.5) | 63.34 | 73.28 | 68.24 | 78.45 | 73.64 | 75.08 | 73.20  | 72.18 |
> | CQG-MBQA (UAE-Large-V1)      | 69.21 | 80.19 | 73.91 | 80.66 | 78.30 | 82.69 | 78.21  | 77.60 |
>
> In the revised version of the paper (to be submitted by the end of November 27), we will incorporate QAEmb-MBQA into **Figure 4** using data from **Tables 1** and **2**.
> Additionally, we will include the ablation study results from **Tables 3** and **4** in the supplementary materials.
> These additions will provide a more comprehensive comparison with the baseline model and further enhance the clarity and depth of our analysis.

---

> ### Author Response · Authors · 2024-11-25
> **Response to Reviewer SSgS [3/3]**
>
> ---
>
> > Implementation details can be moved more to the main paper as they are mostly in appendices.
>
> Thank you for your kind suggestion. We will include additional implementation details in the main paper in the next revision, which will be submitted by the end of November 27.
>
> ---
>
> > In order to achieve performance of regular dense representation models, which aspects of the framework and the implementation details do the authors think worth scaling up? Is there any interesting evidence? e.g., better datasets for generating questions; more questions to form the embedding dimensions, etc..
>
> We carefully selected the training corpus and the number of dimensions to strike a balance between effectiveness and efficiency.
> However, there is room for improvement, particularly through the use of a more diverse and higher-quality training corpus. For domain-specific applications, users can extend the framework by training it on a corpus tailored to their specific domain (no labels required), enabling the generation of questions that are more relevant and discriminative for that context.
> While this extension is a promising direction, it lies beyond the primary focus of this work. We plan to explore this avenue in future research.
>
> ---
>
> > From my understanding in the appendix, the paper uses UAE-large-v1 as the encoding model, why? What happens if the encoding model is some better models - does it help with the performance of the final interpretable embeddings?
>
> We selected `UAE-Large-V1` because it is a state-of-the-art model published in ACL 2024, open-source, highly ranked on the MTEB benchmark, and designed with generalizability in mind, as it does not heavily rely on training data from the MTEB benchmark. This choice aligns with our goal of developing a **general** text embedding framework.
>
> In response to this feedback, we conducted additional experiments, and the results in **Table 4** demonstrate that `UAE-Large-V1` outperformed the other two models, `stella_en_400M_v5` and `gte-large-en-v1.5`.
>
> However, we acknowledge that using an even better encoder could potentially improve final embedding performance by enhancing clustering quality and representation as the backbone of the MBQA model.
> Due to time constraints, we opted to focus on a recently published, high-performing model rather than exhaustively testing all available encoders.
>
> ---
>
> We hope our responses and additional experiments have clarified the points raised and demonstrated the improvements made. Thank you for your valuable feedback, which has significantly strengthened our work.
> We appreciate your time and thoughtful review and look forward to any further suggestions you may have.

---

> > ### Comment · Reviewer_SSgS · 2024-11-26
> >
> > thanks for the response! I am happy to increase the rating. Would love to see the extra experiments to be incorporated in the next version of the paper.

---

> > > ### Author Response · Authors · 2024-11-28
> > >
> > > We sincerely thank the reviewer for the positive feedback on our responses and for increasing the score. The revised paper including the extra experiments has been uploaded, and we deeply appreciate the valuable comments that have significantly contributed to improving the manuscript. Please do not hesitate to let us know if you have any further questions or concerns.

---

### Author Response · Authors · 2024-11-28
**General Response and Revision Summary**

We sincerely thank all reviewers for their insightful and constructive feedback, as well as their kind responses to our clarifications. Your comments have been instrumental in significantly improving the quality of our work.

As acknowledged by the reviewers, our paper addresses an important problem of creating general interpretable semantic text embeddings (sYsh, RC4T, 8TCb, vj78). The proposed CQG-MBQA framework effectively generates discriminative questions for text embeddings (SSgS, sYsh, RC4T) while maintaining cost efficiency (sYsh, RC4T, vj78). The framework is well implemented (SSgS, RC4T, 8TCb) and supported by comprehensive experimental evaluations (SSgS, sYsh, RC4T, vj78). Finally, the evaluation of interpretability and the quality/interpretability tradeoff is well considered (8TCb, vj78).

As promised in our previous responses, we have incorporated the suggested changes and uploaded the revised paper. Changes (excluding minor language edits) are highlighted in blue in the revised paper.

1. **Additional Experiments on Downstream Tasks (Appendix E)**
   We expanded the experimental evaluation to include four additional downstream tasks—classification, pair classification, reranking, and summarization evaluation—in addition to the previously reported STS, retrieval, and clustering tasks.

2. **Ablation Studies on Design Choices (Appendix F)**
   To evaluate the importance of specific design components, we conducted ablation studies on the effects of removing explicit negatives, hard negatives, easy negatives, and the probing mechanism.

3. **Question Filtering for QAEmb Baseline (Appendix G)**
   To ensure a fairer comparison, we introduced LLM-based question filtering for the QAEmb baseline. This clarifies whether CQG’s performance gains stem from the probing mechanism or the generation of higher-quality questions.

4. **Clarifications and Additions**
    * Added discussions on two more related works (S$^3$BERT and LISA) in interpretable embeddings (Lines 124--126 and 129-131 in Section 2).
    * Clarified the model design and task setting of our paper (Lines 138--142 in Section 3).
    * Clarified the contribution of the MBQA model and its connection to existing works (Lines 270--272 in Section 3.2).
    * Added a cost analysis of the CQG method (Lines 297--299 in Section 3.2).
    * Included normalized cognitive load by the number of questions, in addition to the original cognitive load metric (Lines 355--356 in Section 4.1 and Table 4).
    * Added more implementation details of the CQG-MBQA framework and the QAEmb-MBQA baseline in our experiments (Lines 363--375 in Section 4.2 and Appendix D.2).
    * Fixed a minor bug in the evaluation scripts for the NewsSpectrum dataset and updated the results; the claim and overall ranking of models remain unchanged (Table 2).
    * Included the QAEmb-MBQA baseline in Figures 4 and 5.
    * Added a curve of MSE vs. the number of clusters ($k$) to justify the choice of $k$ (Appendix D.1 and Figure 6).

We hope you find these revisions enhance the clarity and robustness of our work. Thank you again for your valuable feedback, and we look forward to any further suggestions.

---

### Meta-Review · Area_Chair_ZtJk · 2024-12-20

**Metareview:**

This paper introduces a framework called CQG-MBQA to create interpretable text embeddings, which automatically produces yes/no questions that capture the differences between texts.

The reviewers recognized the authors’ contributions to their proposed framework for generating discriminative questions while maintaining cost efficiency  (Reviewer SSgS, sYsh, RC4T). Also, the effectiveness of CQG-MBQA was acknowledged by the reviewers for comprehensive experiments across multiple tasks and strong baselines (Reviewer SSgS, sYsh, RC4T, vj78).

Concerns about limited or missing ablation studies on key design choices (e.g., question dimensionality, question difficulty, …) were raised (Reviewer SSgS, RC4T, vj78, 8TCb). In addition, the reviewer sYsh raised concerns about generalizability, specifically whether the method requires generating new sets of yes/no questions for each dataset and if a more universal question set could be designed.

During the rebuttal, the authors made great efforts to address the reviewers’ concerns. For the main concern about lacking ablation studies, the authors added extensive experiments to demonstrate the performance of CQG-MBQA. For the concerns about generalizability, the authors also conducted experiments on four additional downstream tasks during the rebuttal period. Considering all factors, the AC therefore recommends acceptance of this paper.

**Additional Comments On Reviewer Discussion:**

The main concern from Reviewer SSgS, RC4T, vj78, and 8TCb is lacking ablation studies on key design choices. Also, the reviewer sYsh raised concerns about generalizability. Most of the concerns the reviewers mentioned are well addressed by extensive experiments the authors added in the rebuttal period. Taking all points into account, the Area Chair recommends acceptance of this paper.

---

### Decision · Program_Chairs · 2025-01-22

Accept (Poster)